# Efficient Multi-task LLM Quantization and Serving for Multiple LoRA Adapters

**Yifei Xia**
Peking University
yifeixia@stu.pku.edu.cn

**Fangcheng Fu**
Peking University
ccchengff@pku.edu.cn

**Wentao Zhang**
Peking University
wentao.zhang@pku.edu.cn

**Jiawei Jiang**
Wuhan University
jiawei.jiang@whu.edu.cn

**Bin Cui**
Peking University
bin.cui@pku.edu.cn

## Abstract

With the remarkable achievements of large language models (LLMs), the demand for fine-tuning and deploying LLMs in various downstream tasks has garnered widespread interest. Parameter-efficient fine-tuning techniques represented by LoRA and model quantization techniques represented by GPTQ and AWQ are of paramount significance. However, although these techniques have been widely adopted in single-task scenarios, research is scarce in multi-task scenarios. To be specific, we find that mainstream quantization methods would prevent the base LLM from being shared among tasks, so current LLM serving systems are infeasible to integrate LLM quantization with multiple LoRA adapters to achieve memory-efficient multi-task serving. Moreover, existing LLM serving systems lack support for dynamic task addition and overlook the workload differences among tasks, leading to inefficiencies in multi-task scenarios.

This work proposes *LoRA-Inlaid*, an efficient multi-task LLM serving system. On the one hand, *LoRA-Inlaid* designs a flexible and efficient multi-task quantization algorithm (MLGPTQ) that facilitates the sharing of a single quantized model for multiple LoRA adapters, which significantly reduces the memory consumption for model deployment. Meanwhile, it supports adding LoRA adapters for new tasks on the fly, without sacrificing the stability of online services. On the other hand, *LoRA-Inlaid* develops a novel multi-task scheduling algorithm guided by output length prediction and grouping among different tasks, which effectively shrinks the memory consumption and avoids frequent switching of LoRA adapters. Empirical results verify that *LoRA-Inlaid* outperforms existing state-of-the-art LLM serving systems by up to $1.58\times$ in terms of throughput, $1.76\times$ in terms of average latency, $2\times$ in terms of job completion time, and $10\times$ in terms of SLO Attainment, while maintaining the same level of model quality.

## 1 Introduction

Large language models (LLMs) have demonstrated impressive effectiveness in various domains [26, 30, 31, 39], and the demand of deploying LLMs in downstream tasks continues to grow [4, 10,

---

Yifei Xia, Fangcheng Fu, and Bin Cui are with the School of Computer Science and Key Lab of High Confidence Software Technologies (MOE), Peking University. Bin Cui is also with the Institute of Computational Social Science, Peking University (Qingdao). Wentao Zhang is with the Center for Machine Learning Research, Peking University. Jiawei Jiang is with the School of Computer Science, Wuhan University.

38th Conference on Neural Information Processing Systems (NeurIPS 2024).

Table 1: Comparison of supported features of different LLM serving systems

| System | Multi-task Serving | Multi-task Quantization | Dynamic Task Addition | Multi-task Scheduling |
|---|---|---|---|---|
| vLLM [19] & TensorRT-LLM [29] | ✗ | ✗ | ✗ | ✗ |
| S-LoRA [35] & Punica [5] | ✓ | ✗ | ✗ | ✗ |
| *LoRA-Inlaid* (this work) | ✓ | ✓ | ✓ | ✓ |

20, 24, 34, 43, 44, 46, 47]. Given the explosive increase in model size and the limitations of hardware resources, "parameter-efficient fine-tuning" (PEFT) and "quantization-then-deployment" have become the most common pathways for deploying LLMs in downstream tasks [50]. On the one hand, PEFT techniques, represented by LoRA (Low-Rank Adaptation) [16], only train small-scale adapters to adapt the base model to a specific task, significantly reducing the cost of model fine-tuning. On the other hand, low-bit quantization techniques like GPTQ and AWQ [12, 22] can substantially reduce the memory requirements of model deployment and alleviate memory access overhead during inference, while maintaining model quality.

Although mainstream LLM serving systems like vLLM and TensorRT-LLM [19, 29] have integrated support for the quantized deployment of fine-tuned models, these systems focus on single-task serving scenarios. With the rising demand for various downstream tasks, efficiently supporting multi-task servicing scenarios has become increasingly crucial. This has led to the emergence of multi-task serving systems supporting multiple LoRA adapters concurrently, such as S-LoRA and Punica [5, 35]. These systems share a unified base model across different tasks and activate different LoRA adapters based on the incoming requests, enabling the simultaneous processing of multiple tasks in a single batch. However, in multi-task scenarios, existing systems still face three major challenges.

First, existing multi-task serving systems cannot effectively incorporate mainstream model quantization methods such as GPTQ and AWQ. Specifically, these quantization methods require calibration of numerical distributions using task-specific datasets, and the quantization process for each task necessitates activating the corresponding LoRA adapter. Consequently, the base models after quantization are divergent across different tasks, and thus it is infeasible to share a unified quantized model. This limitation leads to performance deficiencies or even unavailability in resource-constrained scenarios.

Second, in practical multi-task serving scenarios, it would be necessary to add new tasks in real time. However, existing systems only support a static number of tasks and are incapable of dynamically adding LoRA adapters. More importantly, after a quantized model is deployed, current solutions do not support any subsequent quantization and deployment for new tasks without affecting the existing tasks. In contrast, adding new tasks typically requires suspending and restarting the serving process, which severely harms the stability and robustness of online services.

Third, incoming requests for different tasks inevitably have workload variations (such as request length, processing time, etc.) and require loading different LoRA adapters for processing. Existing systems overlook these issues during the scheduling for multi-task requests, and thus necessitate loading a large number of adapters in a single scheduling step as well as frequently switching adapters between adjacent scheduling steps, leading to significant efficiency degradation.

To address these challenges, we develop *LoRA-Inlaid*, a resource-efficient and high-performance system for multi-task LLM serving. The main contributions of this paper are as follows.

To begin with, we propose an innovative multi-task quantization algorithm termed MLGPTQ (**M**ulti-**L**oRA **GPTQ**), which utilizes multi-task data to perform joint quantization on the base model. This allows the quantized base model to be shared across multiple tasks. In addition, it supports incremental quantization for newly added tasks without impacting the performance of online services.

Subsequently, we introduce a novel multi-task scheduling strategy based on output length prediction and grouping. This effectively reduces memory consumption and memory swapping overhead in multi-task scenarios, significantly enhancing overall system performance.

Based on these two techniques, we develop a brand new multi-task LLM serving system, namely *LoRA-Inlaid*. As shown in Table 1, *LoRA-Inlaid* integrates multi-task quantization, enables dynamic task addition, and employs the multi-task scheduling strategy, achieving high-performance and flexible multi-task LLM serving in resource-constrained environments.

Finally, experimental results demonstrate that, compared to existing systems, *LoRA-Inlaid* can increase throughput by up to $1.58\times$, reduce average latency and job completion time by up to $1.76\times$ and $2\times$, improve SLO Attainment by up to $10\times$, and support larger-scale language models under the same resource constraints, all while maintaining nearly the same level of model quality.

## 2 Background and Related Works

**Low-Rank Adaptation.** LoRA [16], short for Low-Rank Adaptation, is one of the most widely used parameter-efficient fine-tuning (PEFT) techniques. Unlike full-parameter fine-tuning, LoRA fine-tunes only a small adapter, which consists of much fewer parameters than the base model, significantly reducing the training cost. The key idea behind LoRA is that the fine-tuning process should only introduce small changes to the weight matrix of the base model (denoted by $\mathbf{W} \in \mathbb{R}^{m \times n}$), so we can learn two small, low-rank matrices (denoted by $\mathbf{A} \in \mathbb{R}^{r \times n}, \mathbf{B} \in \mathbb{R}^{m \times r}$ where $r \ll m, n$), and approximate such changes with the product of two matrices (i.e., $\Delta \mathbf{W} \approx \mathbf{BA}$).

**Low-bit Quantization.** Low-bit quantization [7, 8, 12, 22, 23, 42] shrinks the model size effectively and thus reduces the memory requirement when deploying the model. In addition, it usually helps to improve efficiency by decreasing the memory access overhead of the model weights. Consequently, it has been widely adopted in LLM serving. There are various quantization paradigms, with post-training quantization (PTQ) being among the most popular ones. Typically, PTQ computes $X_{\text{INT}} = \texttt{Round}(\alpha\, \texttt{Clip}(X_R/\alpha, Q_{\min}, Q_{\max}))$, where $X_R$ represents the real-valued parameters before quantization, $X_{\text{INT}}$ represents the parameters after quantization to integers, $Q_{min}$ and $Q_{max}$ denote the minimum and maximum values of the quantization range, and $\alpha$ represents the scaling factor. Various PTQ methods calculate the quantization knobs like $\alpha$ with diverse approaches or implement different approximation methods. While mainstream PTQ methods (e.g., GPTQ [12], AWQ [22]) have a common ground that they need to calibrate the numerical distribution based on a small task-specific dataset (a.k.a. *the calibration set*), since numerous studies have revealed the accuracy after quantization with dataset calibration is usually significantly higher than that without dataset calibration [17]. Therefore, this paper focuses on quantization with dataset calibration.

**Scheduling in LLM Serving.** With the explosive applications of LLMs, more and more studies try to evolve the scheduling strategies in LLM serving for better performance. Early systems like FasterTransformer [28] rely on request-level scheduling. Notably, Yu et al. [45] introduced Orca, the first iteration-level scheduling with first-come-first-serve (FCFS) order for better batching. Building on this, mainstream LLM serving systems leverage various batching approaches, such as continuous batching in vLLM [19] and in-flight batching in TensorRT-LLM [29]. FastServe [40] takes the semi-information of requests (e.g., input length, processed time, etc.) into account and tries to minimize average job completion time. However, none of these scheduling strategies consider the characteristics of multi-task scenarios, as we will discuss in §3.3.

**Multi-task Serving Systems.** Since the LoRA fine-tuning technique keeps the base model unaltered, it is feasible to share the same base model across multiple LoRA adapters, so that we can serve requests from multiple tasks within a single batch. Punica [5] and S-LoRA [35] are two notable multi-task serving systems, putting forward the initial efforts to support multi-task LLM serving with multiple LoRA adapters. Specific optimization techniques are proposed. For instance, the Segmented Gather Matrix-Vector (SGMV) kernel is developed to enhance memory and computation efficiency when processing requests from different tasks together. In addition, to allocate more GPU memory to intermediate results (typically, KV cache), existing systems maintain the LoRA adapters in CPU memory and only preserve a relatively small number of LoRA adapters in GPU memory. When a LoRA adapter outside GPU memory is needed, it is necessary to perform memory swapping between the CPU and GPU memory.

## 3 *LoRA-Inlaid*

The overview of *LoRA-Inlaid* is depicted in Figure 1. Given an LLM with multiple LoRA adapters for various downstream tasks, *LoRA-Inlaid* initiates a joint quantization process (§3.1), which produces a unified quantized base model that can be shared across the adapters. During the online serving, if new tasks are to be included on the fly, *LoRA-Inlaid* facilitates a dynamic task addition process (§3.2) that efficiently conducts incremental re-quantization and seamlessly deploys the added tasks.

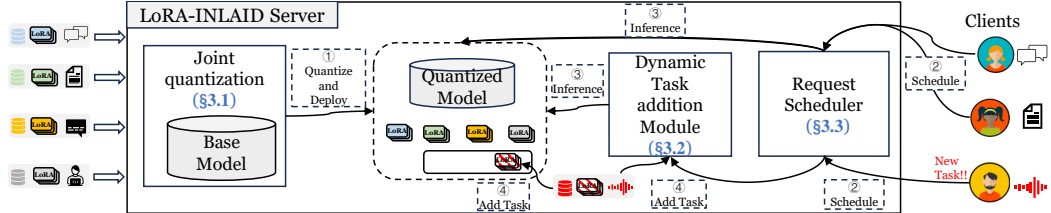

Figure 1: Design overview of *LoRA-Inlaid*. The workflow is labeled with numbers in the diagram. ① *Quantize and Deploy* indicates the initiation of the server performing the multi-task quantization and deploying the quantized model and LoRA online. ② *Schedule* involves utilizing a multi-task scheduling strategy for ③ *Inference*. If a new task is detected, it invokes ④ *Add Task* to dynamically add the new task without interrupting the ongoing services.

Furthermore, *LoRA-Inlaid* employs a multi-task scheduling strategy (§3.3) that takes the workload differences into account for better efficiency.

## 3.1 Multi-task Joint Quantization

As introduced in §2, mainstream quantization methods require task-specific datasets for calibration. In addition, they mostly follow the **Forward-Aggregate Info-Modify Weight-Quant** paradigm in Figure 2. This paradigm first simulates the activation distribution for a given task through **Forward** propagation and **Aggregate**s **Info**rmation of this specific task. Subsequently, it uses the aggregated information to **Modify** model **Weight**s to adapt to the task. Finally, the quantization knobs like scales $\alpha$ are calculated based on the modified weights to **Quant**ize the base model.

However, in multi-task scenarios, since different tasks should provide diverse calibration sets and necessitate unique LoRA adapters for computation, the quantized models of different tasks are inevitably divergent. Intuitively, if we wish to tweak existing quantization methods to make the quantized model shareable across tasks, we should quantize the model without any LoRA adapters. In addition, we should either (i) quantize the model without calibration or (ii) quantize the model with a mixed calibration set consisting of the datasets from all tasks.

However, these approaches fail to accurately capture the unique numerical distribution of each task, and suffer from severe accuracy loss (as evaluated in §4.2). Below we first elaborate on the reason why these approaches fail with the widely used GPTQ [12] and then propose our solution[1].

**Drawbacks of GPTQ in multi-task scenarios.** Directly applying GPTQ in multi-task scenarios has the following drawbacks. First, as aforementioned, GPTQ can only quantize the model without any LoRA adapters, which is infeasible to accurately capture the correct activation information for multiple tasks during **Forward**. Second, in **Aggregate Info**, since the calibration sets from all tasks are mixed, GPTQ simply accumulates the information from different tasks into one Hessian matrix, making each task's specific information diluted and losing the emphasis on critical information from different tasks. Third, in **Modify Weight**, GPTQ relies on the naïve, mix-aggregated Hessian matrix, overlooking the varying importance across tasks, which results in suboptimal outcomes. These drawbacks make the direct application of GPTQ in multi-task scenarios ineffective.

**Our MLGPTQ (Multi-LoRA GPTQ) Algorithm** To address these drawbacks, we propose a multi-task quantization algorithm termed MLGPTQ. Our algorithm enables joint quantization of multiple tasks to retain only one quantized base model, while effectively maintaining the model accuracy by capturing the numerical distributions of all tasks. The goal of MLGPTQ is to minimize the errors of activations before and after quantization, i.e.,

$$\arg\min_{Q(\mathbf{W})} || \sum_{t=1}^{T}((\mathbf{W} + \mathbf{B}_t\mathbf{A}_t)\mathbf{X}_t - (Q(\mathbf{W}) + \mathbf{B}_t\mathbf{A}_t)\mathbf{X}_t)||_2^2, \tag{1}$$

---

[1]Note that the idea of our solution can also be applied to other quantization algorithms which follow the same paradigm (like AWQ) since the drawbacks exist generally. Considering that the choice of backbone algorithm is orthogonal to our goal, we focus on GPTQ in this work due to its widespread adaption, and present how to adapt our solution to AWQ in Appendix D.

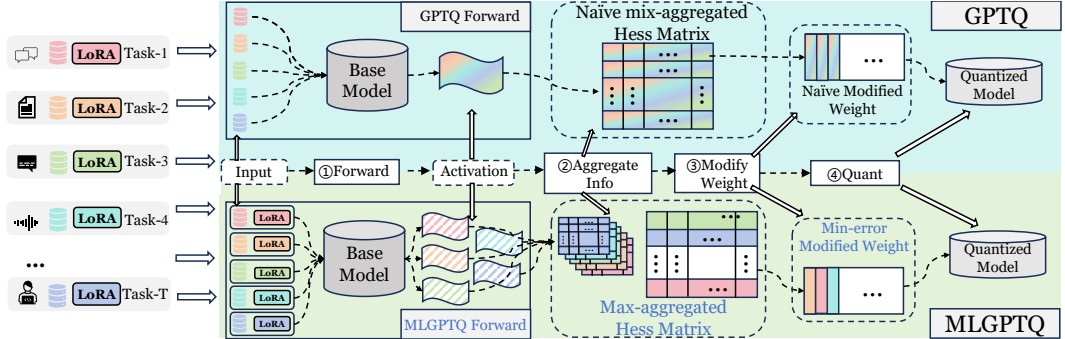

Figure 2: Process of MLGPTQ vs GPTQ. Both MLGPTQ and GPTQ follow the **_Forward-Aggregate Info-Modify Weight-Quant_** paradigm. MLGPTQ primarily improves the first three steps, aiming to better gather and highlight critical information for all tasks.

where $T$ denotes the number of tasks, $\mathbf{A}_t$ and $\mathbf{B}_t$ are the low-rank adapter matrices of the $t$-th task, $\mathbf{X}_t$ is the input of $t$-th task, and $\mathbf{W}$ and $Q(\mathbf{W})$ denote the original and quantized weights of a layer.

As shown in Figure 2, During **_Forward_**, MLGPTQ loads the corresponding LoRA adapters based on each task, accurately computing the activations. In **_Aggregate Info_**, unlike GPTQ's naïve mix-aggregation that disrupts task-specific information, MLGPTQ derives the max-aggregation to solve the objective in Eq. 1 (the derivation can be found in the Appendix A), which has the following form:

$$\nabla \mathbf{W} = -\frac{w_q - Q(w_q)}{(\mathbf{H}_{t^*}^{-1})_{qq}} \mathbf{H}_{t^*}^{-1} e_q, \text{ where } t^* = \underset{t \in \{1, 2, \cdots, T\}}{\arg\max} (\mathbf{H}_t^{-1})_{qq}, \quad (2)$$

where $\mathbf{H}_t$ denotes the Hessian matrix of the $t$-th task, $w_q$ is the $q$-th parameter in $\mathbf{W}$. To be formal, there are primarily two steps in **_Aggregate Info_**. First, it calculates the Hessian matrix information for each task individually (i.e., compute $\{\mathbf{H}_t^{-1}\}_{t=1}^T$) Second, it aggregates the most important information from each one into a max-aggregated Hessian matrix (i.e., $\mathbf{H}_{tmp} = \texttt{MaxAgg}(\{\mathbf{H}_t^{-1}\}_{t=1}^T)$). In **_Modify Weight_**, MLGPTQ utilizes the max-aggregated Hessian matrix to adjust the weights according to Eq. 2. Finally in **_Quant_**, we utilize the modified weights for quantization. Due to space constraints, we only present the core concept of MLGPTQ here. Interested readers are referred to Appendix A for a complete derivation as well as the detailed algorithm.

### 3.2 Dynamic Task Addition

In real-world online services, there is a need for dynamic task addition (i.e., adding new LoRA adapters). In single-task scenarios, adding new tasks typically requires launching more services with extra hardware resources, which does not affect the services for existing tasks. In multi-task scenarios, there would be interference since all tasks share the same base model. However, we find that none of the existing multi-task serving systems address this problem, lacking a proper solution.

Nevertheless, adding new LoRA adapters on the fly in _LoRA-Inlaid_ is inherently far from trivial since the multi-task quantization poses two challenges: _(1. Unseen Distributions)_ Since the MLGPTQ algorithm is invoked before the new tasks are involved, the quantized model has not captured the distribution information about the new tasks, making it infeasible to work with the new LoRA adapters directly. _(2. Serving Interruption)_ Directly re-quantizing the model requires a substantial amount of memory, so it necessitates pausing the ongoing serving for a while to reserve available space for re-quantization, harming the stability of online services. To support dynamic task addition in multi-task scenarios, _LoRA-Inlaid_ tackles these two obstacles, respectively.

To capture the information of new tasks, a naïve solution is to perform full quantization once there are new tasks. Denote $T_1, T_2$ as numbers of existing and new tasks, respectively. The naïve solution runs the two steps of **_Aggregate Info_** above with $T = T_1 + T_2$. However, this leads to redundant computation of $\{\mathbf{H}_t^{-1}\}_{t=1}^{T_1}$. In addition, given the commutative property of the max-aggregation operation, we have $\texttt{MaxAgg}(\{\mathbf{H}_t^{-1}\}_{t=1}^T) = \texttt{MaxAgg}(\texttt{MaxAgg}(\{\mathbf{H}_t^{-1}\}_{t=1}^{T_1}), \texttt{MaxAgg}(\{\mathbf{H}_t^{-1}\}_{t=T_1+1}^{T_2}))$, where the first term $\texttt{MaxAgg}(\{\mathbf{H}_t^{-1}\}_{t=1}^{T_1})$ has already been computed as $\mathbf{H}_{tmp}$ in the previous quantization. Inspired by this, _LoRA-Inlaid_ caches $\mathbf{H}_{tmp}$ so that the incremental quantization can be done as follows. In **_Forward_**, it capture the activation information of new task $T_1 + 1, \cdots, T_2$. In **_Aggregate_**

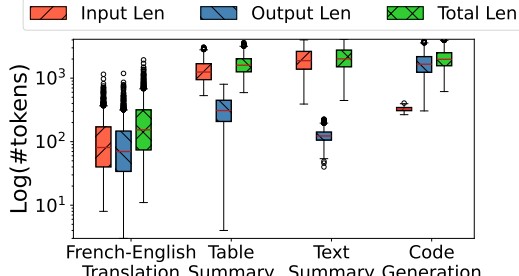

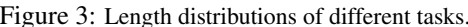

Figure 3: Length distributions of different tasks.

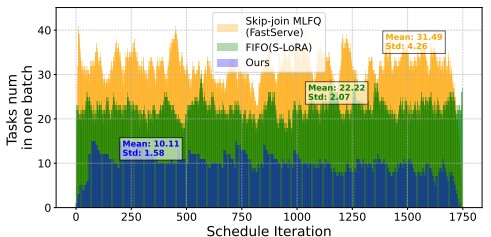

Figure 4: Number of tasks in each scheduling step of different scheduling strategies.

***Info***, it computes the Hessian matrices for new tasks $\{\mathbf{H}_t^{-1}\}_{t=T_1+1}^{T_2}$, and then max-aggregates the $T_2 + 1$ matrices (i.e., $\{\mathbf{H}_t^{-1}\}_{t=T_1+1}^{T_2}$ and the cached $\mathbf{H}_{tmp}^{(cached)}$). At last, it performs ***Modify Weight*** and ***Quant***, as introduced in §3.1. By doing so, incremental quantization with $T_2$ tasks is identical to full quantization with $T_1 + T_2$ tasks, while avoiding redundant computation.

To avoid halting the ongoing services, *LoRA-Inlaid* spawns a background thread for incremental quantization. Moreover, it is done in a layer-by-layer manner to reduce the memory consumption — for each (unquantized) model weight, we load it from CPU memory to GPU memory, perform incremental quantization, remove it from GPU memory, and proceed to the next model weight. The IO between CPU-GPU is overlapped with computation. Thus, *LoRA-Inlaid* supports seamless task addition on the fly and has very little influence on the ongoing services, as evaluated in §4.4.

Putting them together, *LoRA-Inlaid* develops an asynchronous, layer-wise re-quantization mechanism, which accomplishes incremental quantization with the new tasks and cached Hessian matrices asynchronously, without interrupting the serving.

### 3.3 Multi-task Scheduling

Despite extensive research on scheduling strategies for LLM serving, these approaches primarily focus on single-task serving, leaving the unique characteristics in the multi-task scenarios neglected. Below we analyze two limitations of existing scheduling strategies in multi-task serving. Besides, due to the space constraint, we briefly introduce the corresponding solutions in *LoRA-Inlaid*, while leaving the details of our multi-task scheduling algorithm in Appendix B.

*Limitation 1: Divergent Output Length Distributions Leading to High Average Completion Time.* As shown in Figure 3, the distributions of input and output lengths vary significantly across different tasks, while requests of the same task exhibit clustering effects. Current strategies mainly rely on semi-information (e.g., input length, processed time, etc.) to make the scheduling decisions, but do not consider the information of output length since it is not the prior knowledge. Intuitively, this may work fine for single-task scenarios where the vast majority of requests fall within the same workload and thus the clustering effect exists. However, it is unsuitable for multi-task scenarios due to the divergent output length distributions across different tasks. Eventually, we find that existing scheduling strategies suffer from heavy performance degradation when applied to multi-task serving.

*Solution 1: Scheduling Guided by Output Length Prediction.* Existing research has shown that the output lengths can be accurately predicted by a small, distilled model given the requests [49]. Inspired by this, we leverage a number of small models, to predict the output lengths of incoming requests. Particularly, upon receiving a new request, we predict its output length on CPU using a small model (255MB). Note that the output length prediction takes about 16 milliseconds for one request on CPU, while it takes about 200 milliseconds or more to finish the inference of one request on GPU. Hence, we can completely overlap the prediction, without occupying any GPU computing resources. Based on the predictions, we employ a Shortest Remaining Time First (SRTF) scheduling, which prioritizes requests with the shortest remaining processing time and has been proven to minimize the average completion time in the field of job scheduling [37].

*Limitation 2: Excessive Tasks Involved in each Step Leading to Expensive Memory Access Overhead.* Due to the randomness and dynamicity of request arrivals, multiple tasks are to be scheduled in each step. However, owing to the lack of consideration upon the task for each request, existing scheduling

Table 2: Model quality of different approaches under different tasks (std-dev given in parentheses). GPTQ and AWQ are in gray background color since the quantized models produced by them cannot be shared across different tasks. We mark the best multi-task quantization approaches (i.e., the best among MLGPTQ, GPTQ$_{tweaked}$, AWQ$_{tweaked}$, and RTN) in bold.

| Dataset | trans-fr | trans-cs | trans-id | trans-nl | trans-da | trans-sw | QTsum | xlsum | tiny-codes | GSM8k | med-qa | malicious |
|---|---|---|---|---|---|---|---|---|---|---|---|---|
| Metric | S_BLEU | S_BLEU | S_BLEU | S_BLEU | S_BLEU | S_BLEU | ROUGE-1 | ROUGE-1 | ROUGE-1 | Acc (%) | Acc (%) | Acc (%) |
| Unquantized | 34.45 (0.01) | 31.89 (0.02) | 33.94 (0.01) | 30.94 (0.01) | 35.04 (0) | 31.14 (0.01) | 49.38 (0) | 41.28 (0.01) | 31.72 (0.02) | 32.14 (0) | 80.9 (0) | 37.44 (0) |
| MLGPTQ (4-bit) | **34.05 (0.02)** | **31.16 (0.02)** | **33.63 (0.01)** | **30.73 (0.04)** | **34.39 (0.01)** | **31.20 (0.01)** | **49.02 (0)** | **40.96 (0.02)** | **30.85 (0.05)** | **31.62 (0)** | **76.7 (0.68)** | **36.44 (0.6)** |
| GPTQ$_{tweaked}$ (4-bit) | 33.91 (0.02) | 28.95 (0.21) | 32.88 (0.08) | 30.48 (0.07) | 33.47 (0.04) | 28.94 (0.06) | 48.23 (0.02) | 39.77 (0.06) | 29.25 (0.10) | 31.51 (0) | 74.81 (2.53) | 35.05 (1.15) |
| AWQ$_{tweaked}$ (4-bit) | 33.88 (0.04) | 29.45 (0.11) | 33.01 (0.06) | 29.99 (0.09) | 33.34 (0.11) | 30.11 (0.07) | 47.96 (0.03) | 40.12 (0.08) | 30.23 (0.07) | 30.51 (0.01) | 75.42 (1.13) | 35.68 (0.33) |
| RTN (4-bit) | 33.79 (0) | 29.64 (0.01) | 32.96 (0.01) | 30.33 (0) | 33.96 (0) | 30.46 (0.02) | 47.54 (0.01) | 40.27 (0.02) | 30.63 (0.02) | 31.01 (0) | 76.15 (0) | 33.78 (0) |
| GPTQ (4-bit) | 34.07 (0.02) | 31.19 (0.03) | 33.79 (0.02) | 30.86 (0.15) | 34.57 (0.02) | 31.08 (0.08) | 49.26 (0.02) | 40.89 (0.06) | 30.92 (0.06) | 31.35 (0) | 76.7 (2.66) | 36.25 (0.44) |
| AWQ (4-bit) | 34.17 (0.03) | 31.19 (0.05) | 33.72 (0.07) | 30.69 (0.08) | 34.21 (0.08) | 31.07 (0) | 49.04 (0.12) | 41.10 (0.02) | 31.03 (0.04) | 31.45 (0) | 75.42 (1.26) | 36.18 (0.28) |
| MLGPTQ (3-bit) | **31.72 (0.39)** | **26.93 (0.58)** | **30.11 (0.63)** | **27.97 (1.04)** | **30.77 (0.50)** | **28.06 (0.53)** | **47 (0.38)** | **39.07 (0.22)** | **27.62 (0.47)** | **28.74 (0)** | **54.84 (7.94)** | **31.90 (0.47)** |
| GPTQ$_{tweaked}$ (3-bit) | 31.3 (0.62) | 25.89 (0.7) | 28.18 (0.75) | 23.54 (1.01) | 24.09 (0.51) | 21.12 (0.39) | 45.99 (0.26) | 38.32 (0.15) | 23.80 (0.46) | 28.30 (0) | 54.02 (7.9) | 30.93 (0.24) |
| AWQ$_{tweaked}$ (3-bit) | 31.57 (0.94) | 26.45 (0.59) | 25.13 (0.62) | 24.46 (1.02) | 26.77 (0.7) | 19.79 (1.02) | 45.13 (0.17) | 37.62 (0.46) | 21.83 (0.46) | 28.24 (0) | 53.66 (19) | 31.03 (0.34) |
| RTN (3-bit) | 26.02 (0.01) | 0.03 (0) | 0.03 (0) | 0.06 (0) | 0.05 (0) | 0.05 (0) | 0.9 (0) | 0.10 (0) | 0.34 (0) | 26.38 (0) | 51.3 (0) | 31.4 (0) |
| GPTQ (3-bit) | 30.83 (0.31) | 26.19 (0.58) | 31.88 (0.65) | 28.21 (0.81) | 32.93 (0.26) | 29.75 (0.3) | 47.22 (0.17) | 39.53 (0.17) | 26.12 (0.11) | 28.16 (0) | 56.03 (12.39) | 31.26 (0.5) |
| AWQ (3-bit) | 31.23 (0.63) | 25.35 (0.18) | 30.35 (0.73) | 28.65 (1.15) | 31.23 (0.32) | 28.77 (0.62) | 47.13 (0.17) | 39.23 (0.44) | 27.16 (0.57) | 28.63 (0) | 53.66 (4.08) | 31.77 (0.29) |

strategies typically involve a great number of tasks, and the system has to load lots of LoRA adapters in each step, as shown in Figure 4. As it is well known that the decoding phase of LLM inference is usually bounded by the memory bandwidth, the need for loading more LoRA adapters further exacerbates this issue. Worst still, we observe that the sets of involved tasks vary significantly between consecutive steps. As introduced in § 2, multi-task serving systems only preserve limited GPU memory space for LoRA adapters, and swap them between CPU and GPU memory when necessary. Consequently, existing scheduling strategies force the system to frequently swap LoRA adapters, rendering performance degradation due to the expensive memory swapping overhead.

*Solution 2: Reducing Tasks Involved via Grouping.* To address the limitation, we adopt a simple yet effective grouping-based approach, which partitions requests into groups according to their tasks to guide scheduling. On one hand, to avoid involving excessive tasks in each step, we set a grouping coefficient $\beta$ (10 by default) and keep the number of involved tasks below $\beta$ in each step. On the other hand, to alleviate the memory swapping overhead, we prioritize tasks involved in the previous step, aiming to use the LoRA adapters for more consecutive steps. In addition, we maintain a starvation queue based on the waiting time to get rid of starvation, striking a good balance among the tasks.

## 4 Experiments

### 4.1 Experimental Setup

**Hardware Environment.** All experiments are conducted on one RTX 4090 GPU or one RTX 3090 GPU, with GPU memory capacity of 24GB. Detailed specifications can be found in Appendix C.1.

**Datasets and Workloads.** For accuracy tests, we consider 12 tasks in total, including six translation tasks [38] (trans-fr, trans-cs, trans-id, trans-nl, trans-da, trans-sw), one text summarization task [14] (xlsum), one table summarization task [48] (QTsum), one code generation task [13] (tiny-codes), one math QA task [6] (GSM8k), one medical QA task [2] (med-qa), and one malicious detection task [1] (malicious). Detailed descriptions are provided in Appendix C.2. For efficiency tests, we follow prior works [19, 35] to generate different levels of request rates using the Gamma process.

**Models.** We conduct experiments with LLaMA2-7B and LLaMA2-13B [25]. For accuracy tests, open-source fine-tuned models are used, as detailed in Appendix C.2. For efficiency tests, following S-LoRA [35], we consider LoRA adapters with different ranks (8, 16, 32, 64) across the served tasks to simulate diverse serving scenarios.

**Metrics.** For the accuracy test, we focus on SacreBLEU (S_BLEU) [18], ROUGE [21] and Accuracy (Acc). Details of these metrics are provided in Appendix C.3. For efficiency tests, the considered metrics include throughput, average request latency, job completion time (JCT), and SLO (service level objective) Attainment [33] (the percentage of requests completed within the expected latency). By default, we test the serving for 1 minute, with an expected latency of 6 seconds.

**Baselines.** For accuracy tests, we focus on two kinds of baselines introduced in §3.1, i.e., (i) the floating-point round-to-nearest quantization without calibration (denoted as RTN), and (ii) GPTQ [12] and AWQ [22] with a mixed calibration set from all tasks (denoted as GPTQ$_{tweaked}$ and AWQ$_{tweaked}$).

---

Source code is available at https://github.com/PKU-DAIR/LoRA_Inlaid.

To be fair, these baselines quantize the base model without any LoRA adapters to ensure the quantized model is shareable. Besides, to facilitate the comparison to single-task quantization, we further consider GPTQ and AWQ for each task individually with the corresponding calibration set and LoRA adapter (denoted as GPTQ and AWQ), which cannot generate a shareable quantized model though.

For efficiency tests, we compare *LoRA-Inlaid* with vLLM [19] and S-LoRA [35]. For vLLM, we quantize each model into 4-bit and launch multiple processes (each with one quantized model) on the same GPU to achieve multi-task serving. For S-LoRA, which does not support deploying quantized models, we deploy one half-precision (16-bit) model and let multiple LoRA adapters share it. For *LoRA-Inlaid*, we use 4-bit quantization in all efficiency tests. Note that although the model is quantized, the computation during inference is still executed in half-precision (i.e., the model weights are dequantized before computation).

## 4.2 Model Quality after Quantization

We first assess the model quality after quantization of different approaches. Table 2 presents the model quality of different tasks under different metrics. It can be seen that all quantization methods incur accuracy drops compared to no quantization. For 4-bit quantization, the average accuracy drops for MLGPTQ, GPTQ$_{tweaked}$, AWQ$_{tweaked}$, and RTN are 1.70%, 4.72%, 4.50%, 4.02%, respectively. For 3-bit quantization, MLGPTQ consistently achieves the best results, outperforming GPTQ$_{tweaked}$ and AWQ$_{tweaked}$ by 59.30%, 69.98% in average. While RTN suffers from up to 74.41% accuracy drops. In addition, for GPTQ and AWQ, which could not produce a shareable model, the average accuracy drops are very close to MLGPTQ (e.g., 1.00%, 1.89% respectively for 4-bit quantization). This proves that our work achieves comparable model quality against single-task quantization while enabling multi-task sharing after quantization.

We also conduct experiments to anatomize the effectiveness of MLGPTQ. To be more comprehensive, five additional metrics (ROUGE1, ROUGE2 [21], NIST_MT [9], METEOR [9], and G_BLEU [41], detailed in Appendix C.3) for evaluating machine translation quality are included. In addition, we further consider a variant of MLGPTQ termed MLGPTQ$_{no\_target}$, which intentionally excludes the calibration set and LoRA adapter of a target task during quantization (the other tasks are not affected).

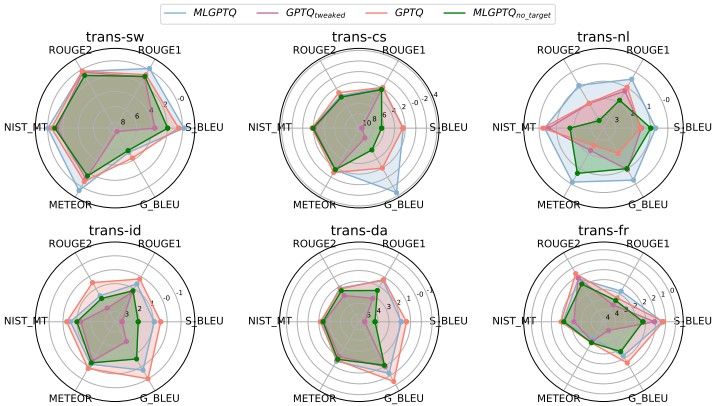

Figure 5: Effectiveness anatomy. The radar charts show the relative accuracy drops compared to no quantization (outer is better).

According to the results in Figure 5, we point out that the effectiveness of MLGPTQ stems from two factors: ① whether the information (e.g., activation and Hessian matrix) of each task is correctly captured, and ② whether the data distribution of each task is involved during the forward pass of quantization. In particular, GPTQ$_{tweaked}$ (missing ①) and MLGPTQ$_{no\_target}$ (missing ②) exhibit higher accuracy drops compared to MLGPTQ and GPTQ (both fulfilling ① and ②) in almost all metrics. These results verify that the design of MLGPTQ fits multi-task quantization well.

## 4.3 End-to-end System Performance

**Throughput, Latency, and JCT.** We evaluate the system performance with various numbers of tasks and request arrival rates. The throughput, latency, and JCT are shown in Figure 6. Overall, *LoRA-Inlaid* consistently outperforms S-LoRA and vLLM. S-LoRA fails to serve LLaMA2-13B due to the lack of support for deploying quantized models. Since vLLM necessitates launching multiple processes to achieve multi-task serving, the memory consumption grows linearly w.r.t. the number

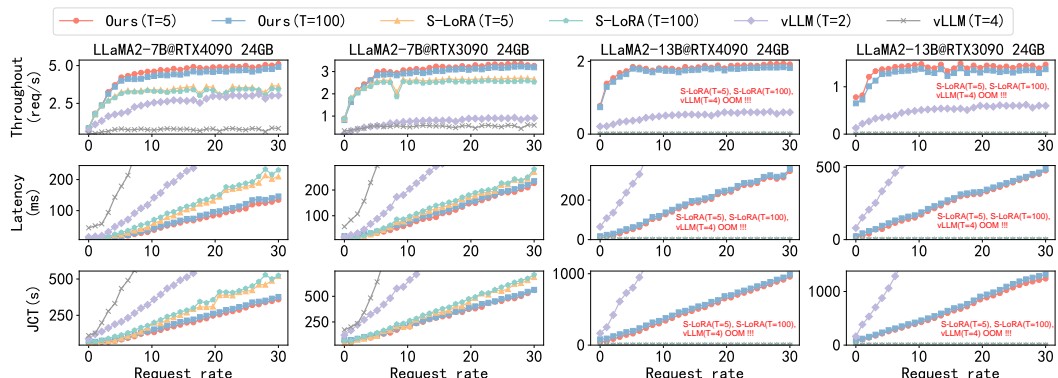

Figure 6: System performance in terms of throughput (higher is better), latency (lower is better), and JCT (lower is better) under various request rates ($x$-axis) and numbers of tasks ($T$).

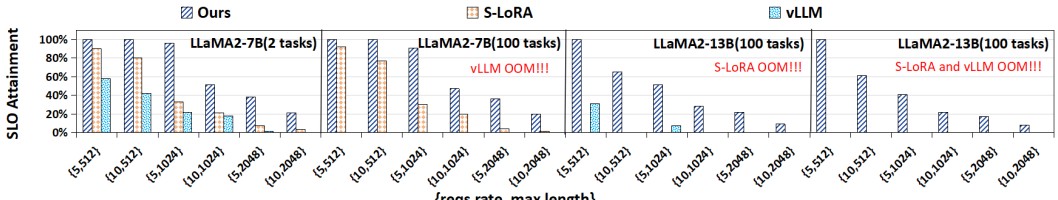

Figure 7: SLO Attainment (higher is better) under various serving loads (RTX 4090).

Table 3: Scalability comparison in terms of throughput (reqs/s, higher is better) under different request rates and number of LoRA adapters (LLaMA2-7B@RTX 4090).

| Task num | 2 | | | 3 | | | 4 | | | 5 | | | ... | 100 | | | 1000 | | |
|---|---|---|---|---|---|---|---|---|---|---|---|---|---|---|---|---|---|---|---|
| Reqs rate | 5 | 10 | 20 | 5 | 10 | 20 | 5 | 10 | 20 | 5 | 10 | 20 | ... | 5 | 10 | 20 | 5 | 10 | 20 |
| *LoRA-Inlaid* | 3.89 | 4.70 | 4.86 | 3.78 | 4.66 | 4.81 | 3.82 | 4.77 | 4.89 | 3.71 | 4.61 | 4.73 | ... | 3.60 | 4.25 | 4.58 | 3.42 | 4.02 | 4.22 |
| S-LoRA | 2.93 | 3.45 | 3.51 | 2.97 | 3.38 | 3.54 | 2.91 | 3.36 | 3.58 | 2.97 | 3.40 | 3.55 | ... | 2.87 | 3.35 | 3.36 | 2.78 | 3.26 | 3.28 |
| vLLM | 1.77 | 2.46 | 2.98 | 1.02 | 1.68 | 2.27 | 0.77 | 0.76 | 0.80 | OOM | OOM | OOM | ... | OOM | OOM | OOM | OOM | OOM | OOM |

of tasks, and it encounters out-of-memory (OOM) errors in several cases. In contrast, *LoRA-Inlaid* supports all cases well. More importantly, since *LoRA-Inlaid* is able to reserve more memory for intermediate results (e.g., KV cache) in serving, it achieves higher performance than the baselines. For instance, *LoRA-Inlaid* surpasses S-LoRA by 26.5%, 31.3%, 24.1% on average, and up to 58.1%, 76.3%, and 99.9%, in terms of throughput, latency, and JCT, respectively.

**SLO Attainment.** We also assess the SLO Attainment under different serving loads by varying the request rates and maximum request lengths. The results are shown in Figure 7. In short, compared to S-LoRA and vLLM, *LoRA-Inlaid* improves the SLO Attainment by 3.9×, 8.5× on average, and up to 10×, 38×, respectively. Furthermore, we observe that as the request rate or maximum sequence length increases, S-LoRA and vLLM experience a steep decline in SLO Attainment while *LoRA-Inlaid* does not. This demonstrates the excellent adaptability of *LoRA-Inlaid* to various serving loads.

### 4.4 More Experiments

**Scalability.** We investigate the scalability w.r.t. number of tasks. As shown in Table 3, vLLM suffers from significant performance decline, dropping by 56%-73% when the number of tasks increases from 2 to 4, and eventually encountering out-of-memory (OOM) errors when the number of tasks reaches 5. In contrast, under all experimented request rates, the throughput of *LoRA-Inlaid* hardly declines, even with 1000 tasks served simultaneously. S-LoRA also supports a large number of tasks, while *LoRA-Inlaid* consistently achieves better performance across all kinds of workloads.

**Ablation Studies of Multi-task Scheduling and Multi-task Quantization.** We compare different scheduling strategies on *LoRA-Inlaid*. The results are shown in the left of Figure 8. "Ours (w/o group)", "Ours (w/o prediction)" and "Ours (w/o SRTF)" represent three variants of our multi-task scheduling strategy without task grouping, without output length prediction and without the prediction-based SRTF, respectively. "FIFO" is the strategy adopted in S-LoRA and vLLM, and "Skip-join

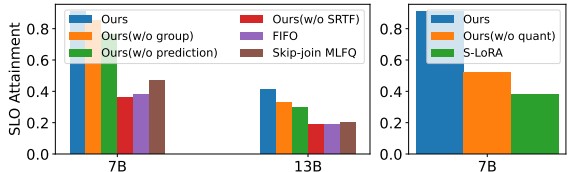
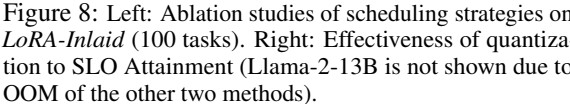
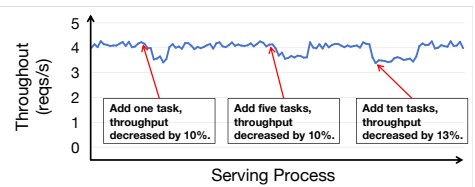

Figure 8: Left: Ablation studies of scheduling strategies on *LoRA-Inlaid* (100 tasks). Right: Effectiveness of quantization to SLO Attainment (Llama-2-13B is not shown due to OOM of the other two methods).

Figure 9: Impact of dynamic task addition on online throughput (LLaMA2-7B@RTX 4090 with 100 tasks initially, and the request rate is 30 reqs/s).

MLFQ" represents the strategy in FastServe [40]. It is evident that our multi-task scheduling strategy achieves the best performance in terms of SLO Attainment. The designs of task grouping, output length prediction, and SRTF increase the SLO Attainment by 1.16×, 1.23× and 2.27× on average, respectively. We also explore the individual impact of multi-task quantization as shown in the right of Figure 8. Specifically, we consider a variant of *LoRA-Inlaid*, which disables quantization (i.e., the served model is not quantized), denoted as "Ours (w/o quant)". The results show that multi-task quantization brings 39% improvement ("Ours" vs. "Ours (w/o quant)") when serving the 7B model. Additionally, without quantization, it will lead to OOM when serving the 13B model.

**Dynamic Task Addition.** We evaluate the ability of dynamic task addition in *LoRA-Inlaid* by adding 1, 5, and 10 tasks to a heavily loaded service on the fly. The results in Figure 9 show that the throughput undergoes 10%-13% of degradation during the task addition, regardless of the number of tasks added. This is worthwhile given

|  | Forward+Cal Hess Matrix | Agg Hess matrix +Quant | Total | Peak Memory |
|---|---|---|---|---|
| **Full Quant** | 1403(±21)s | 415(±6)s | 1818(±22)s | 9.2GB |
| **Incr Quant (offline)** | 663(±11)s | 416(±5)s | 1079(±12)s | 9.2GB |
| **Incr Quant** | 889(±11)s | 469(±6)s | 1358(±13)s | 2.5GB |

Table 4: Time cost of quantization.

that the online service need not be interrupted. Meanwhile, to evaluate the time consumption of dynamic task addition, we conducted an experiment where there are 5 tasks in the ongoing service and another 5 tasks need to be added. We measured the time cost of three approaches: "Full Quant", which halts the serving and performs full quantization with 10 tasks, "Incr Quant offline", an offline variant (which halts the serving) of our incremental quantization on the 5 new tasks without layer-by-layer quantization, and "Incr Quant", our incremental quantization with the 5 new tasks, which works concurrently with the ongoing service. As shown in Table 4, by avoiding the redundant computation, the time cost of forward process and calculation of Hess matrix can be reduced greatly, accelerating quantization. Moreover, although the layer-by-layer mechanism slows down the quantization by 1.26 × due to the extra IO, it reduces the memory greatly and does not halt the serving. These empirical results validate the flexibility and robustness of *LoRA-Inlaid* for multi-task serving.

## 5 Conclusion and Limitations

In this work, we focused on LLM serving in multi-task scenarios and developed a multi-LoRA task serving system, namely *LoRA-Inlaid*. On one hand, we designed a flexible and efficient dynamic multi-task quantization algorithm that supports the joint quantization of models for multiple tasks, significantly reducing the memory requirements for model deployment. We also facilitated real-time dynamic task addition, enhancing the stability and flexibility of online services. On the other hand, we introduced a novel multi-task scheduling strategy based on output length prediction and grouping, effectively resolving the issues of high memory overhead and frequent memory swapping when applying existing strategies in multi-task scenarios. Extensive experiments demonstrated that *LoRA-Inlaid* significantly outperforms existing LLM serving systems.

Despite the effectiveness of *LoRA-Inlaid*, it still has several limitations. First, our quantization does not detect the existence of malicious or poisoning tasks, which might be intentionally crafted to harm the other tasks. Second, our scheduling does not consider the fairness among tasks (e.g., balancing the total numbers of output tokens for all tasks), which may be essential for shared service platforms. Third, it only supports language tasks while requiring some system re-designs for multi-modal tasks. We wish to leave the exploration of these issues as future works.

## Acknowledgments and Disclosure of Funding

This work is supported by National Natural Science Foundation of China (U22B2037, U23B2048), China National Postdoctoral Program for Innovative Talents (BX20230012), China Postdoctoral Science Foundation (2024M750103), Beijing Natural Science Foundation (4244080), research grant No. SH-2024JK29, the Fund of Kunpeng and Ascend Center of Excellence (Peking University), and High-performance Computing Platform of Peking University. Fangcheng Fu and Bin Cui are the corresponding authors.

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

# A Details of MLGPTQ

## A.1 Derivation of MLGPTQ

MLGPTQ is based on GPTQ, adapting the corresponding LoRA adapter according to the activation values of the input data to minimize the error in activation values before and after quantization. For only one task, GPTQ aims to solve the following problem:

$$\arg\min_{Q(\mathbf{W}_l)} ||\mathbf{W}_l\mathbf{X}_l - Q(\mathbf{W}_l)\mathbf{X}_l||_2^2, \tag{3}$$

where $\mathbf{W}_l$ represents the weight of $l$-th layer in the base model, $Q(\mathbf{W}_l)$ represents the quantized version of $\mathbf{W}_l$, $\mathbf{X}_l$ is the input of $l$-th layer. In other words, the objective of GPTQ is to find a $Q(\mathbf{W}_l)$ for each layer's weight $\mathbf{W}_l$ through layer-wise quantization, in order to minimize the changes in activation values. Since we are discussing quantization within a single layer, we will omit the subscript $l$ for simplicity in the rest of this section.

As proved by [11], solving Eq. 3 can be transformed into solving Eq. 4 as follows.

$$\arg\min_{Q(\mathbf{W})} E := \sum_{i=1}^{d_{\text{row}}} ||\mathbf{W}_{i,:}\mathbf{X} - Q(\mathbf{W})_{i,:}\mathbf{X}||_2^2. \tag{4}$$

Since the model has converged through training before quantization, existing works generally assume the model has reached a local minimum. Thus, when we add a small adjustment $\nabla\mathbf{W}$ to the parameter $\mathbf{W}$, according to Taylor expansion, we have

$$\nabla E = \left(\frac{\partial E}{\partial \mathbf{W}}\right)^T \nabla\mathbf{W} + \frac{1}{2}\nabla\mathbf{W}^T \cdot \mathbf{H} \cdot \nabla\mathbf{W} + O(||\nabla\mathbf{W}||^3), \tag{5}$$

where $\mathbf{H} = \partial^2 E/\partial\mathbf{W}^2$ represents the Hessian matrix. Again, since the model has converged, existing works generally assume its first-order partial derivative is close to zero and thus negligible. By neglecting the first-order partial derivative and higher-order terms, we have

$$\nabla E = \frac{1}{2}\nabla\mathbf{W}^T \cdot \mathbf{H} \cdot \nabla\mathbf{W}. \tag{6}$$

Recall that our goal is to quantize $\mathbf{W}$ to $Q(\mathbf{W})$. Denote $\nabla w_q = Q(w_q) - w_q$, where $w_q$ represents the $q$-th element of $\mathbf{W}$. Then, the problem to solve can be re-written as

$$\arg\min_q \left\{ \arg\min_{\nabla\mathbf{W}} \left(\frac{1}{2}\nabla\mathbf{W}^T \cdot \mathbf{H} \cdot \nabla\mathbf{W}\right) \left[e_q^T\nabla\mathbf{W} + w_q = Q(w_q)\right] \right\}, \tag{7}$$

where $e_q$ represents a unit vector with a value of 1 at position $q$ and 0 elsewhere. Since this is a constrained convex optimization problem, based on the method of Lagrange multipliers, it is necessary to solve the following equation:

$$L = \frac{1}{2}\nabla\mathbf{W}^T \cdot \mathbf{H} \cdot \nabla\mathbf{W} + \lambda(e_q^T\nabla\mathbf{W} + w_q - Q(w_q)). \tag{8}$$

By taking the partial derivatives of $\nabla W$ and $\lambda$, and setting them to zero to find the steady-state solution, we have

$$\begin{cases} \frac{1}{2}(\mathbf{H} + \mathbf{H}^T)\nabla\mathbf{W} + \lambda e_q = 0 \\ e_q^T\nabla\mathbf{W} + w_q - Q(w_q) = 0 \end{cases} \tag{9}$$

Solving this, we get

$$\lambda = \frac{w_q - Q(w_q)}{(\mathbf{H}^{-1})_{qq}}, \tag{10}$$

$$\nabla\mathbf{W} = -\frac{w_q - Q(w_q)}{(\mathbf{H}^{-1})_{qq}}\mathbf{H}^{-1}e_q, \tag{11}$$

$$\nabla E = \frac{(w_q - Q(w_q))^2}{2(\mathbf{H}^{-1})_{qq}}, \tag{12}$$

where $(\mathbf{H}^{-1})_{qq}$ represents the value at the diagonal position $(q, q)$ of $\mathbf{H}^{-1}$, which is the inverse of the Hessian matrix.

For the Hessian matrix, we say that $\mathbf{H} = 2\mathbf{X}\mathbf{X}^T$ by proving the following Lemma.

**Lemma A.1.** *Let $a$ be a $1 \times n$ row vector and $X$ be an $n \times m$ matrix. The Hessian matrix of the quadratic form $\|aX\|_2^2$ is $2XX^T$.*

*Proof.* Let $a = [a_1, a_2, \ldots, a_n]$ be a $1 \times n$ row vector, and let $X = [x_{ij}]$ be an $n \times m$ matrix.

Let's denote $y = aX$. Then, $y$ is a $1 \times m$ row vector with elements $y_j$ defined as

$$y_j = \sum_{i=1}^{n} a_i x_{ij}.$$

Thus, we have

$$\|aX\|_2^2 = (aX) \cdot (aX)^T = \sum_{j=1}^{m} y_j^2 = \sum_{j=1}^{m} \left( \sum_{i=1}^{n} a_i x_{ij} \right)^2.$$

We can expand this expression and re-write it as a quadratic form, i.e.,

$$\|aX\|_2^2 = \sum_{j=1}^{m} \sum_{i=1}^{n} \sum_{k=1}^{n} a_i a_k x_{ij} x_{kj}.$$

To find the Hessian matrix of this quadratic form, we treat it as a quadratic form in $a$. Let $Q$ be the coefficient matrix of this quadratic form.

$$\|aX\|_2^2 = aQa^T.$$

The $(i, k)$ element of $Q$ is given by

$$Q_{ik} = \sum_{j=1}^{m} x_{ij} x_{kj}.$$

Thus, the matrix $Q$ can be written as

$$Q = XX^T.$$

And the Hessian matrix is twice $Q$:

$$H = 2Q = 2XX^T.$$

Therefore, the Hessian matrix of $\|aX\|_2^2$ is $2XX^T$, which completes the proof. $\square$

GPTQ quantizes the weight $\mathbf{W}$ row by row. For each row, according to Eq. 12 mentioned above, it finds the minimum $w_q$ that leads to an increase in the loss function due to quantization, then calculates scales via $\alpha = \frac{max(\mathbf{W}_{i,:}) - max(\mathbf{W}_{i,:})}{Q_{max}}$, performs quantization using $\alpha$, and finally update the remaining values using Eq. 11. This process repeats until all parameters have been updated.

MLGPTQ considers the scenario of quantization for multiple tasks. During the forward propagation process, it dynamically loads the corresponding LoRA adapter for each task to simulate the correct activation values for the respective tasks. Consequently, the problem we need to solve is as follows.

$$\underset{Q(\mathbf{W})}{\arg\min} || \sum_{t=1}^{T} ((\mathbf{W} + \mathbf{B}_t \mathbf{A}_t) \mathbf{X}_t - (Q(\mathbf{W}) + \mathbf{B}_t \mathbf{A}_t) \mathbf{X}_t) ||_2^2, \tag{13}$$

where $T$ denotes the number of tasks, $\mathbf{A}_t$ and $\mathbf{B}_t$ are the low-rank adapter matrices of the $t$-th task, $\mathbf{X}_t$ is the input of $t$-th task, and $\mathbf{W}$ and $Q(\mathbf{W})$ denote the original and quantized weights of a layer.

Denote $\widetilde{\mathbf{W}}_t := \mathbf{W} + \mathbf{B}_t \mathbf{A}_t$, then the problem is re-written as

$$\underset{Q(\mathbf{W})}{\arg\min} \sum_{i=1}^{d_{\text{row}}} \left\| \sum_{t=1}^{T} (\widetilde{\mathbf{W}}_{\mathbf{t}\,i,:} \mathbf{X}_t - Q(\widetilde{\mathbf{W}}_{\mathbf{t}})_{i,:} \mathbf{X}_t) \right\|_2^2. \tag{14}$$

Same as GPTQ, we could get $T$ Hessian matrix $\mathbf{H}_t$, where $t \in [1, T]$. To minimize the objective function, we can obtain the updating formulas for $\mathbf{W}$ and $E$ in MLGPTQ:

$$\nabla \mathbf{W} = -\frac{w_q - Q(w_q)}{(\mathbf{H}_{t^*}^{-1})_{qq}} \mathbf{H}_{t^*}^{-1} e_q, \nabla E = \frac{(w_q - Q(w_q))^2}{2\left((\mathbf{H}_{t^*}^{-1})_{qq}\right)}, \text{ where } t^* = \underset{t \in [1,T]}{\arg\max}\left((\mathbf{H}_t^{-1})_{qq}\right). \quad (15)$$

The updating method described here leads to the max-aggregation method proposed in § 3.1, which always selects the Hessian matrix of the task that minimizes $\nabla E$ for updating, ultimately reducing the overall error and thus better-guiding parameter updates.

### A.2 Pseudocode of MLGPTQ

We provide the pseudocode of MLGPTQ in Algorithm 1, and we also highlight the differences compared to directly applying GPTQ to multi-task scenarios, which is termed GPTQ$_{tweaked}$ in our work (i.e., with a mixed calibration set and without any LoRA adapters).

Due to the high complexity and numerical instability of the process described in Appendix A.1, we leverage the following optimizations to accelerate the quantization, partly inspired by the practical implementation of GPTQ [12].

**Random Order Optimization.** GPTQ requires updating weights in the order that produces the smallest quantization error $\nabla E$. For $\mathbf{W} \in \mathbb{R}^{m \times n}$, the complexity of GPTQ is $O(mn^3)$. However, using a random order achieves similar results and facilitates GPU parallel optimization [12].

**Batch Processing.** Since weight updates between different columns of the same matrix $\mathbf{W}$ are non-redundant, we use batch processing and delayed updates, with 128 columns processed at a time, to enhance computation speed.

**Cholesky Decomposition.** Using numerically stable Cholesky decomposition to pre-compute the necessary information increases computational stability.

---

**Algorithm 1** Routines of  MLGPTQ  and  GPTQ$_{tweaked}$  to quantize one layer in multi-task scenarios.

---

**Input:** $\{\mathbf{X}_t\}_{t=1}^T$                 ▷ The inputs of different tasks of this layer

1:   $\mathbf{X}_t \leftarrow (\mathbf{W} + \mathbf{B}_t \mathbf{A}_t)\mathbf{X}_t$     $\mathbf{X}_{mixed} \leftarrow \sum_1^T \mathbf{W}\mathbf{X}_t$         ▷ Forward pass

2:   $\mathbf{H}_t^{-1} \leftarrow \text{Cholesky}((2\mathbf{X}_t\mathbf{X}_t^T + \lambda \mathbf{I}))^T$, for $\forall t \in [1, T]$

    $\mathbf{H}^{-1} \leftarrow \sum_1^T \text{Cholesky}((2\mathbf{X}_{mixed}\mathbf{X}_{mixed}^T + \lambda \mathbf{I})^{-1})^T$       ▷ Cal inv of Hessian matrix

3:   $\mathbf{H}_{tmp} \leftarrow \text{zero\_like}(\mathbf{H}_t^{-1})$              ▷ Init the tmp Hessian matrix

4:   $\mathbf{Q} \leftarrow 0_{d_{\text{row}} \times d_{\text{col}}}$                       ▷ Store quantized results

5:   $\mathbf{E} \leftarrow 0_{d_{\text{row}} \times B}$                        ▷ Store quantization errors in blocks

6: **for** $i \leftarrow 0, B, 2B, \ldots$ **do**

7:      **for** $j \leftarrow i, \ldots, i + B - 1$ **do**

8:          $\mathbf{Q}_{:,j} \leftarrow \text{quant}(\mathbf{W}_{:,j})$           ▷ Column-wise quantization

9:          $\mathbf{E}_{:,j-i} \leftarrow (\mathbf{W}_{:,j} - \mathbf{Q}_{:,j}) / \max_{t \in [1,T]}\left((\mathbf{H}_t^{-1})_{jj}\right)$

          $\mathbf{E}_{:,j-i} \leftarrow (\mathbf{W}_{:,j} - \mathbf{Q}_{:,j}) / (\mathbf{H}^{-1})_{jj}$       ▷ Update quantization error

10:        $(\mathbf{H}_{tmp})_{:,s} \leftarrow (\mathbf{H}_{t^*}^{-1})_{:,s}$ where $t^* = \arg\max_{t \in [1,T]}\left((\mathbf{H}_t^{-1})_{ss}\right)$, for $s \in [j, i + B - 1]$

          $\mathbf{W}_{:,j:(i+B)} \leftarrow \mathbf{W}_{:,j:(i+B)} - \mathbf{E}_{:,j-i} \cdot (\mathbf{H}_{tmp})_{:,j:(i+B)}$

          $\mathbf{W}_{:,j:(i+B)} \leftarrow \mathbf{W}_{:,j:(i+B)} - \mathbf{E}_{:,j-i} \cdot (\mathbf{H}^{-1})_{:,j:(i+B)}$    ▷ Update weights in current block

11:      **end for**

12:     $\mathbf{W}_{:,(i+B)} \leftarrow \mathbf{W}_{:,(i+B)} - \mathbf{E} \cdot (\mathbf{H}_{tmp})_{i:(i+B),(i+B)}$

     $\mathbf{W}_{:,(i+B)} \leftarrow \mathbf{W}_{:,(i+B)} - \mathbf{E} \cdot (\mathbf{H}^{-1})_{i:(i+B),(i+B)}$    ▷ Update weights in remaining blocks

13: **end for**

---

# B  Details of our Multi-task Scheduling Strategy

Our multi-task scheduling strategy is based on task grouping and prediction-based SRTF. Compared to other scheduling strategies, it is well-suited for multi-task scenarios, achieving excellent results. The pseudocode is shown in Algorithm 2, with the helper functions $generate\_new\_batch$ and $schedule\_new\_batch$ in Algorithm 3 and 4. There are four queues maintaining different requests in our system: ① $prefill\_reqs$: requests that have not yet entered the prefill stage (i.e., new requests that have not yet been served). ② $decoding\_reqs$: requests in the decoding stage. ③ $hungry\_prefill\_reqs$: requests in a starving state that have not yet been served. ④ $hungry\_decoding\_reqs$: requests in a starving state in the decoding stage.

First, we check if $running\_batch$ is empty (line 7) to determine whether the system is in a cold start phase (i.e., the previous scheduling step did not have any requests to serve). If yes, we perform a cold start by calling $generate\_new\_batch$ to get a new batch to schedule from $prefill\_reqs$ (line 8). If $new\_batch$ is not empty, we proceed to $prefill$ with $new\_batch$ (line 10-11). Otherwise, the system remains idle since there are no requests (line 13). If $running\_batch$ is not empty, we check if we have consecutively executed $max\_cont\_decode$ decoding steps (line 16). If yes, we call $generate\_new\_batch$ to schedule new requests from $prefill\_reqs$ and $hungry\_prefill\_reqs$ (line 17-20). Otherwise, we check if the continuous scheduling for a batch has reached a pre-defined threshold $max\_cont\_decode\_one\_batch$ (line 22). If yes, we call $schedule\_new\_batch$ to schedule new requests in the decoding stage from $decoding\_reqs$ and $hungry\_decoding\_reqs$ according to the SRTF and grouping strategy (line 23-25). Otherwise, we continue processing the current $running\_batch$ (line 27-28). This two-level threshold strategy effectively reduces the frequent swapping of LoRA adapters and KV caches due to frequent batch switching. Additionally, by adjusting the threshold size, we can ensure the immediacy and flexibility of scheduling.

---

**Algorithm 2** Our multi-task scheduling strategy based on grouping and SRTF.

---

**Input:** Four queues: $prefill\_reqs$, $decoding\_reqs$, $hungry\_prefill\_reqs$, $hungry\_decoding\_reqs$

```
 1: running_batch ← ∅
 2: max_cont_decode ← Threshold value for decoding
 3: max_cont_decode_one_batch ← Threshold value for decoding one batch
 4: decode_count ← 0
 5: decode_count_one_batch ← 0
 6: while not terminated do
 7:     if running_batch is empty then
 8:         new_batch ← generate_new_batch(prefill_reqs)
 9:         if new_batch ≠ ∅ then
10:             Perform prefill(new_batch)
11:             decoding_reqs ← decoding_reqs + new_batch
12:         else
13:             Keep idle
14:         end if
15:     else
16:         if decode_count ≥ max_cont_decode then
17:             new_batch ← generate_new_batch(prefill_reqs, hungry_prefill_reqs)
18:             Perform prefill(new_batch)
19:             decoding_reqs ← decoding_reqs + new_batch
20:             decode_count ← 0
21:         else
22:             if decode_count_one_batch ≥ max_cont_decode_one_batch then
23:                 new_batch ← schedule_new_batch(decoding_reqs, hungry_decoding_reqs)
24:                 Perform decode(running_batch)
25:                 decode_count_one_batch ← 0
26:             else
27:                 Perform decode(running_batch)
28:                 decode_count_one_batch ← decode_count_one_batch + 1
29:             end if
30:             decode_count ← decode_count + 1
31:         end if
32:     end if
33: end while
```

---

---

**Algorithm 3** The utility function $generate\_new\_batch$ for Algorithm 2.

---

1: **function** GENERATE_NEW_BATCH($prefill\_reqs$, $hungry\_prefill\_reqs$)
2:     Sort $prefill\_reqs$ by $len(prompt) + predict\_output\_len$ ascending
3:     Sort $hungry\_prefill\_reqs$ by $waiting\_time$ descending, then $len(prompt) + predict\_output\_len$ ascending
4:     $new\_batch \leftarrow \emptyset$
5:     **for** each $req$ in $hungry\_prefill\_reqs$ **do**
6:         **if** can_add_req($req$, $new\_batch$) and meet_max_lora($req$, $new\_batch$) **then**
7:             $new\_batch$.append($req$); $hungry\_prefill\_reqs$.remove($req$)
8:         **end if**
9:     **end for**
10:     **for** each $req$ in $prefill\_reqs$ **do**
11:         **if** can_add_req($req$, $new\_batch$) and meet_max_lora($req$, $new\_batch$) **then**
12:             $new\_batch$.append($req$); $prefill\_reqs$.remove($req$)
13:         **end if**
14:     **end for**
15:     **if** $new\_batch$ not full **then**
16:         **for** each $req$ in $hungry\_prefill\_reqs + prefill\_reqs$ **do**
17:             **if** can_add_req($req$, $new\_batch$) **then**
18:                 $new\_batch$.append($req$); $prefill\_reqs$.remove($req$)
19:             **end if**
20:         **end for**
21:     **end if**
22:     **for** each $req$ in $prefill\_reqs$ and $hungry\_prefill\_reqs$ **do**
23:         $req.waiting\_time \leftarrow req.waiting\_time + 1$
24:     **end for**
25:     **for** each $req$ in $prefill\_reqs$ **do**
26:         **if** $req.waiting\_time \geq$ Threshold **then**
27:             $prefill\_reqs$.remove($req$); $hungry\_prefill\_reqs$.append($req$)
28:         **end if**
29:     **end for**
30:     **return** $new\_batch$
31: **end function**

---

---

**Algorithm 4** The utility function $schedule\_new\_batch$ for Algorithm 2.

---

1: **function** SCHEDULE_NEW_BATCH($running\_batch$, $decoding\_reqs$, $hungry\_decoding\_reqs$)
2:     Sort $decoding\_reqs$ by $predict\_output\_len - len(output)$ ascending
3:     Sort $hungry\_decoding\_reqs$ by $waiting\_time$ descending, then $predict\_output\_len - len(output)$ ascending
4:     $new\_batch \leftarrow \emptyset$
5:     **for** each $req$ in $hungry\_decoding\_reqs$ **do**
6:         **if** can_add_req($req$, $new\_batch$) and $req.lora \in running\_batch$ **then**
7:             $new\_batch$.append($req$); $hungry\_decoding\_reqs$.remove($req$)
8:         **end if**
9:     **end for**
10:     **for** each $req$ in $decoding\_reqs$ **do**
11:         **if** can_add_req($req$, $new\_batch$) and $req.lora \in running\_batch$ **then**
12:             $new\_batch$.append($req$); $decoding\_reqs$.remove($req$)
13:         **end if**
14:     **end for**
15:     **if** $new\_batch$ not full **then**
16:         **for** each $req$ in $hungry\_decoding\_reqs + decoding\_reqs$ **do**
17:             **if** can_add_req($req$, $new\_batch$) **then**
18:                 $new\_batch$.append($req$); $decoding\_reqs$.remove($req$)
19:             **end if**
20:         **end for**
21:     **end if**
22:     **for** each $req$ in $decoding\_reqs$ **do**
23:         $req.waiting\_time \leftarrow req.waiting\_time + 1$
24:     **end for**
25:     **return** $new\_batch$
26: **end function**

---

# C  More Experimental Details

## C.1  Experimental Environment

The GPU platforms for evaluation are shown in Table 5. The RTX 3090 platform is equipped with Intel(R) Core(TM) i9-10900X CPU @ 3.70GHz and 256GB host memory, while the RTX 4090 platform is equipped with Intel(R) Xeon(R) Gold 6330 CPU @ 2.00GH and 512GB host memory.

Table 5: Experimental GPU platforms in detail.

| Platform | RTX 4090 | RTX 3090 |
|---|---|---|
| GPU cores | 16384 | 10496 |
| GFLOPS | 51640 | 35580 |
| Memory Capacity | 24 GB | 24 GB |
| Memory Access Bandwidth | 1024 GB/s | 936 GB/s |

## C.2  Summary of Evaluated Tasks

We primarily select 12 major datasets for testing, covering tasks such as machine translation, text summarization, table summarization, code generation, math QA, medical QA and malicious detection. For each task, we use open-source models from Hugging Face[2] that have been fine-tuned using the training set of the corresponding dataset and evaluated on the test set. The summary is presented in Table 6.

Table 6: Dataset Summary

| Dataset Name | Abbreviation | Avg. Input Length | Avg. Output Length | Type of Task |
|---|---|---|---|---|
| OPUS-French-English | trans-fr | 121 | 105 | Machine Translation |
| OPUS-Czech-English | trans-cs | 47 | 47 | Machine Translation |
| OPUS-Indonesian-English | trans-id | 47 | 38 | Machine Translation |
| OPUS-Vietnamese-English | trans-nl | 72 | 65 | Machine Translation |
| OPUS-Danish-English | trans-da | 72 | 71 | Machine Translation |
| OPUS-Swedish-English | trans-sw | 64 | 65 | Machine Translation |
| XLSum | xlsum | 2595 | 125 | Text Summarization |
| QTSUMM | QTsum | 1350 | 339 | Table Summarization |
| tiny-codes | tiny-codes | 328 | 1890 | Code Generation |
| gsm8k | GSM8k | 240 | 293 | Math Question Answer |
| medical_meadow_mmmlu | med-qa | 367 | 1 | Medical Question Answer |
| malicious-600k | malicious | 59 | 1 | Malicious Detection |

For the machine translation tasks of different languages, we consider the classic bilingual translation dataset OPUS [38], specifically choosing six translation tasks: French-to-English, Czech-to-English, Indonesian-to-English, Vietnamese-to-English, Danish-to-English, and Swedish-to-English, considering the diversity of languages.

For the text translation task, we consider the XLSum dataset [14], a comprehensive and diverse dataset comprising 1.35 million professionally annotated text-summary pairs extracted from the BBC.

For the table summarization task, we consider the QTSumm dataset [48], which is a large-scale dataset for query-centric summarization tasks on tabular data. It contains 7,111 human-annotated query-summary pairs and 2,934 tables covering various topics.

For the code generation task, we consider the tiny-codes dataset [13]. This dataset consists of 16 million short and clear code snippets, aiding LLM models in learning how to reason using both natural and programming languages. The dataset includes a variety of programming languages such as Python, TypeScript, JavaScript, Ruby, Julia, Rust, C++, Bash, Java, C#, and Go.

---

[2]https://huggingface.co/kaichuup

For the math QA task, we choose the GSM8k (Grade School Math 8K) dataset [6], which consists of 8.5K high quality linguistically diverse grade school math word problems. The dataset was created to support the task of question answering on basic mathematical problems that require multi-step reasoning.

For the medical QA task, we choose medical_meadow_mmmlu [2], which contains 3.79k medical multiple choice question.

For malicious detection task, we choose malicious-600k [1], which contains 641k URLs to determine whether they are malicious.

### C.3 Summary of Metrics

(1) **SacreBLEU** [18], represented as S_BLEU, is a classic machine translation evaluation standard. It scores by comparing the n-gram overlap between the machine translation output and one or more reference translations, while also considering a brevity penalty to prevent favoring overly short translation outputs.

(2) **rouge1** [21], represented as ROUGE1, calculates the overlap ratio of words between the machine-generated text and the reference text, i.e., the unigram (single word) match rate.

(3) **rouge2** [21], represented as ROUGE2, similar to rouge1, indicates the bigram (two-word sequence) match rate.

(4) **nist_mt** [9], represented as NIST_MT, is an improvement on BLEU that gives higher weight to less common words, encouraging diversity and accuracy in translation.

(5) **meteor** [3], abbreviated as METEOR, calculates the score based on the harmonic mean of precision and recall and introduces a penalty factor to penalize excessive mismatches.

(6) **google_bleu** [41], abbreviated as G_BLEU, is an improvement on BLEU that adjusts for smoothing methods, n-gram weights, and brevity penalties to optimize its performance.

# D   Multi-LoRA AWQ Migration

**Algorithm 5** Routines of ☐MLAWQ☐ and ☐AWQ*tweaked*☐ to quantize on layer in multi-task scenarios.

**Input:** $\{\mathbf{X}^t\}_{t=1}^T$, ratio_search_space ▷ The inputs of different tasks of this layer

1: $\mathbf{M}^t \leftarrow (\mathbf{W} + \mathbf{B}^t\mathbf{A}^t)\mathbf{X}^t$    $\mathbf{M}_{\text{mixed}} \leftarrow \sum_1^T \mathbf{W}\mathbf{X}^t$    ▷ Forward to get monitoring matrix

2: $\mathbf{W}_{\text{mean}} \leftarrow \mathbf{W}.\text{mean}(0)$

3: $\mathbf{X}^t_{\text{mean}} \leftarrow \mathbf{X}^t.\text{mean}(0)$ for $\forall t \in [1, T]$    $\mathbf{X}_{\text{mean}} \leftarrow \{\mathbf{X}^t\}_{t=1}^T.\text{mean}(0)$    ▷ Aggregate Info: Per-channel
    mean of $\mathbf{X}$

4: best_s $\leftarrow$ None

5: min_errors $\leftarrow \infty \cdot \mathbf{1}_{\dim(X^t.\text{shape[-1]})}$    min_error $\leftarrow \infty$    ▷ Initialize minimum error(s)

6: **for** ratio **in** ratio_search_space **do**    ▷ Aggregate Info: Calculate best_s

7:    $s_t \leftarrow \left( \frac{\mathbf{X}^t_{\text{mean}}.\text{pow(ratio)}}{\mathbf{W}_{\text{mean}}.\text{pow(1-ratio)}} \right)$ for $\forall t \in [1, T]$    $s \leftarrow \left( \frac{\mathbf{X}_{\text{mean}}.\text{pow(ratio)}}{\mathbf{W}_{\text{mean}}.\text{pow(1-ratio)}} \right)$    ▷ Calculate $s$

8:    $\mathbf{W}^t_{\text{scaled}} \leftarrow \mathbf{W} \cdot s_t$    $\mathbf{W}_{\text{scaled}} \leftarrow \mathbf{W} \cdot s$    ▷ Scale $\mathbf{W}$

9:    $\mathbf{X}^t_{\text{scaled}} \leftarrow \mathbf{X}^t / s_t$    $\mathbf{X}_{\text{scaled}} \leftarrow \{\mathbf{X}^t\}_{t=1}^T / s$    ▷ Scale $\mathbf{X}$

10:    $\text{errs}_t \leftarrow \|\mathbf{M}^t - (\alpha\left(\text{round}\left(\text{clamp}\left(\mathbf{W}^t_{\text{scaled}}/\alpha, \text{min\_val}, \text{max\_val}\right)\right)\right) + \mathbf{B}^t\mathbf{A}^t)\mathbf{X}^t_{\text{scaled}}\|_{2,\text{columns}}$

     $\text{err} \leftarrow \|\mathbf{M}_{\text{mixed}} - \alpha\left(\text{round}\left(\text{clamp}\left(\mathbf{W}_{\text{scaled}}/\alpha, \text{min\_val}, \text{max\_val}\right)\right)\right)\mathbf{X}_{\text{scaled}}\|$   ▷ Use pseudo quantized
    W run forward to cal quant error of this ratio, where $\alpha$ is the scale factor of pseudo quant

11:    $\text{min\_errs}[j], \text{best\_s}[j] \leftarrow \min_t(\text{errs}_t[j]), s_{\arg\min_t(\text{errs}_t[j])}[j] \;\; \forall j \in \mathbf{X}^t.\text{shape[-1]}, \forall t \in [1, T]$

     $\text{min\_err} \leftarrow \min(\text{min\_err}, \text{err}), \quad \text{best\_s} \leftarrow (\text{err} < \text{min\_err}) \,?\, s \,:\, \text{best\_s}$
                                          ▷ Aggregate the min error to get best_s

12: **end for**

13: $\mathbf{W}_{\text{modified}} \leftarrow \mathbf{W} \cdot \text{best\_s}$                         ▷ Modify weight

14: $\mathbf{W}_{\text{quant}} \leftarrow \text{quantize}(\mathbf{W}_{\text{modified}})$              ▷ Quantize modified weight

15: **Return** $\mathbf{W}_{\text{quant}}$

Our work centers on GPTQ due to its widespread use, but our solution can also adapt to AWQ. We presented the differences between MLAWQ and AWQ_tweaked in Alg 5. As introduced in §3.1, most quantization methods follow the **Forward-Aggregate Info-Modify Weight-Quant** paradigm. In essence, AWQ_tweaked smooths outliers by multiplying weights with a smoothing factor, best_s, to minimize per-channel quant error:

- In **Forward**, the input is multiplied by the weights to create an unquantized monitor, guiding min error quantization (line 1).
- In **Aggregate Info**, the average of all samples and weights is calculated for each channel (lines 2&3) to determine smoothing factor $s$ (line 7). $\mathbf{W}$ and $\mathbf{X}$ are smoothed to remove outliers (lines 8&9). Then, smoothed $\mathbf{W}$ is pseudo-quantized (quantize-then-dequantize to simulate round loss) and compared to the unquantized monitor for quantization error (line 10). This process iterates over various ratios (line 6), selecting the factor with the smallest error as best_s (line 11). Then, this best_s is used to **Modify the weight** (line 12), followed by the **Quant** (line 13) process using the modified weight.

The drawbacks discussed in §3.1 also exist for AWQ_tweaked in multi-task quantization:

- **Forward** (line 1): It can't pass LoRA adapters during activation distribution simulation, causing quantization bias during inference.
- **Aggregate Info**: It uses $X_{\text{mean}} = X.\text{mean}(0)$, a naive mixed average of multi-tasks' info. Since each task affects each channel differently, simply averaging blurs distributions, ignoring individual effects.

As explained in §3.1 and shown in Fig 2, MLGPTQ mainly improves the first three steps to tackle GPTQ's issues in multi-task scenarios. Similarly, we can fix AWQ's issues to create a better multi-task quantization algorithm MLAWQ:

- **Forward**: MLAWQ loads corresponding LoRA adapter for each task to participate in forward propagation, accurately simulating real activation distribution (line 1).

- **Aggregate Info**: Instead of mixing and averaging features of each column across all tasks to compute $s$, MLAWQ computes the average for each task separately to get $s_i$ (line 3&7). Then it calculates quantization error for each column rather than the entire matrix (line 10). If the $i$-th task results in the smallest quantization error for the $j$-th column, it sets best_s $[j] = s_i[j]$ (line 11). This approach allows optimal error minimization, showing each task's individual effect on different channels, enhancing **Aggregate Info** (lines 3&6-11), and improving the **Modify Weight** (line 12) and **Quant** (line 13) processes.

In summary, our work identifies common drawbacks of current single-task quantization methods in multi-task scenarios. By addressing these issues, we can develop more precise multi-task quantization algorithms.

# E  Broader Impact and Future Work

This work proposes *LoRA-Inlaid*, a brand new LLM serving system for the multi-task scenario. *LoRA-Inlaid* is featured with a series of innovations, specifically the multi-task joint quantization algorithm, the dynamic task addition mechanism, and the multi-task scheduling strategy. Considering the booming research and applications of LLMs in various downstream tasks, we believe *LoRA-Inlaid* has the potential to gain widespread adoption and shed light on the high-performance and resource-efficient system designs for follow-up works in the field of LLM serving. However, there are several issues that *LoRA-Inlaid* does not consider currently.

On the one hand, the multi-task quantization algorithm in *LoRA-Inlaid* does not involve the detection of malicious or poisoning tasks that may bring negative impacts on the other tasks. One of the most typical use cases of *LoRA-Inlaid* is the personalization of LLMs, where clients can upload their data to create personalized LoRA adapters using the same base model (or directly upload their self-tuned LoRA adapters). The server is responsible for serving requests from all these clients using the proposed *LoRA-Inlaid* system. Fortunately, these LoRA adapters are independently manufactured, so we can apply malicious detection to them individually. For instance, the server can prepare a rich set of evaluations to assess the security risks of each LoRA adapter, including violence, discrimination, unlawful responses, etc. If any LoRA adapters fail to pass the evaluation, the server can reject serving them.

On the other hand, the multi-task scheduling strategy in *LoRA-Inlaid* ignores the fairness among different tasks (e.g., controlling the number of output tokens to be close), which may make our work ineffective under some settings. To measure fairness among tasks, we can compute a weighted combination of numbers of input and output tokens for each task. This is because the prefilling and decoding phases in LLM inference have different workload characteristics [15](also why these tokens differ in online service pricing). Then, we can borrow the idea of Weighted Fair Queueing (WFQ) [32] for scheduling different tasks. We wish to address these issues in the future.

