# OpenReview forum: "Efficient Multi-task LLM Quantization and Serving for Multiple LoRA Adapters"
_NeurIPS.cc/2024/Conference — NeurIPS 2024 poster_

### Official Review · Reviewer_gmzG · 2024-07-03

**Soundness:** 3
**Presentation:** 3
**Contribution:** 3
**Rating:** 7
**Confidence:** 4

**Summary:**

This paper mainly focuses on the quantization problem of large language models and the problem of low-rank decomposition solvers, and proposes a method for quantizing large language models for multiple tasks and integrating multiple low-rank decomposition solvers. The article first analyzes that the current mainstream quantization method will make it impossible to share model parameters among tasks when processing multiple tasks. Therefore, the current LLM service system cannot integrate LLM quantization with multiple LoRA solvers to achieve memory-efficient multi-task support. At the same time, the existing LLM service system lacks support for dynamic task addition. Therefore, the LoRA-Inlaid proposed in this paper designs a flexible and efficient multi-task quantization algorithm, which is convenient for multiple LoRA solvers to share a single quantization model, greatly reducing the memory consumption of model deployment. On the other hand, LoRA-Inlaid develops a new multi-task scheduling algorithm based on output length prediction and grouping, which effectively reduces memory consumption and avoids frequent switching of LoRA solvers. This innovative multi-task quantization algorithm MLGPTQ uses multi-task data to perform joint quantization on the basic model. This allows the basic model of quantization to be shared between multiple tasks. In addition, it also supports incremental quantization of newly added tasks without affecting the performance of the service.

**Strengths:**

This paper proposes an innovative multi-task LLM quantization method for large language models, which can be used without introducing any other memory calculation consumption.
It handles multiple quantification tasks with the following main advantages:
1. This innovative multi-task quantization algorithm MLGPTQ uses multi-task data to perform joint quantification on the basic model. This allows the quantified base model to be shared across multiple tasks.
2. Supports incremental quantification of adding new tasks, which improves the defect that most current systems can only support a constant number of tasks. Tasks can be added dynamically without having a great impact on the current task process. No need Pause or restart the current service process.
3. A multi-task scheduling strategy based on output length prediction and grouping is proposed. This effectively reduces memory consumption and memory swapping overhead in multi-tasking scenarios, significantly improving the overall performance of the system.
4. LoRA-Inlaid integrates multi-task quantification, realizes dynamic task addition, and adopts a multi-task scheduling strategy to achieve high-performance and flexible computing services for multi-task LLM in an environment with limited resources.

**Weaknesses:**

This article is a very excellent and innovative algorithmic article for quantitative model improvement. The algorithm process is clearly explained, the process is clear, and the proof process is very detailed. The proof ideas are clear and complete. However, the following small issues can be considered:
1. The symbols of the proof process in this article need to be further sorted out, for example, does the q-th parameter refer to the qth parameter or the qth row parameter?
2. This article mentions that the Lagrange multiplier method needs to be used to solve max-aggregation, but the solution process, including the final result, will have multiple matrix inversions. Then the condition number of the matrix needs to be further considered to see if it will affect the accuracy and speed of the solution.
3. Figure 2 mentions the difference between the method in this article and the GPTQ method, which is very clear, but this article also proposes other quantitative methods. Can you explain why GPTQ is simply compared with MLGPTQ, or the AWQ method is also represented in this way to show the highlights of this method?
4. This article has less or less explanation of the multi-task scheduling strategy. It is recommended that the author add more comparative experiments on this module, otherwise the content is relatively lacking as an innovation point.
5. For large model tasks, there may be some priorities among the tasks. So based on the multi-task scheduling method in this article, how should we consider the situation where there are multiple tasks and the task levels have an order?

**Questions:**

This article is a very excellent and innovative algorithmic article for quantitative model improvement. The algorithm process is clearly explained, the process is clear, and the proof process is very detailed. The proof ideas are clear and complete. However, the following small issues can be considered:
1. The symbols of the proof process in this article need to be further sorted out, for example, does the q-th parameter refer to the qth parameter or the qth row parameter?
2. This article mentions that the Lagrange multiplier method needs to be used to solve max-aggregation, but the solution process, including the final result, will have multiple matrix inversions. Then the condition number of the matrix needs to be further considered to see if it will affect the accuracy and speed of the solution.
3. Figure 2 mentions the difference between the method in this article and the GPTQ method, which is very clear, but this article also proposes other quantitative methods. Can you explain why GPTQ is simply compared with MLGPTQ, or the AWQ method is also represented in this way to show the highlights of this method?
4. This article has less or less explanation of the multi-task scheduling strategy. It is recommended that the author add more comparative experiments on this module, otherwise the content is relatively lacking as an innovation point.
5. For large model tasks, there may be some priorities among the tasks. So based on the multi-task scheduling method in this article, how should we consider the situation where there are multiple tasks and the task levels have an order?

**Limitations:**

This article mentions possible limitations of this method at the end. The first is that the quantitative method in this article does not detect the existence of malicious or poisoned tasks, which may be intentionally used to harm other tasks. Second, our scheduling does not consider fairness among tasks, which may be essential for shared service platforms.
Third, it only supports language tasks, while requiring some system redesign for multimodal tasks. In addition to the three possible limitations mentioned in the article, the priority of tasks and the order between tasks may also be points that need to be considered.
Secondly, the calculation time of the intermediate process of MLGPTQ may vary significantly depending on the model. In addition, if some tasks may need to be exited midway during the training process, how should the model proposed in this article handle this situation.

---

> ### Author Rebuttal · Authors · 2024-08-07
>
> ## Q1
> It refers to the $q$-th parameter, i.e., $w_q$ denotes the $q$-th element of $\mathbf{W}$ after it is flattened.
>
> ## Q2
> In Appendix A.2, we implement MLGPTQ using **Cholesky decomposition** to increase speed and computational stability, similar to GPTQ's implementation (https://github.com/AutoGPTQ/AutoGPTQ/blob/v0.7.0/auto_gptq/quantization/gptq.py#L117). To address the comment, we record the condition numbers of the Hessian matrices for different layers, providing the mean and std-dev below. Results show that condition numbers are consistently within $10^3$, which is a reasonable value range in practice.
>
> | | trans-fr | trans-cs | trans-id | trans-nl | trans-da | trans-sw | QTsum | xlsum | tiny-codes |
> | ----- | ------ | -------- | -------- | -------- | -------- | -------- | ----- | ----- | ---------- |
> |mean| 285.1    | 316.3    | 345.7    | 321.0    | 318.3    | 324.1    | 258.3 | 203.2 | 232.9     |
> |std-dev| 101.0    | 108.7    | 94.8     | 107.9    | 108.2    | 103.9    | 92.2  | 92.9  | 89.5       |
>
> ## Q3
> Our work centers on GPTQ due to its widespread use, but our solution can also adapt to AWQ. We presented the differences between MLAWQ and $AWQ_{tweaked}$ in _Alg A_ of the one-page PDF (similar to Alg 1 of our manuscript). As introduced in Section 3.1, most quantization methods follow the **_Forward-Aggregate Info-Modify Weight-Quant_** paradigm. In essence, $AWQ_{tweaked}$ smooths outliers by multiplying weights with a smoothing factor, best_s, to minimize per-channel quant error:
>
> - In **_Forward_**, the input is multiplied by the weights to create an unquantized monitor, guiding min error quantization (line 1).
> - ​In **_Aggregate Info_**, the average of all samples and weights is calculated for each channel (lines 2&3) to determine smoothing factor s (line 7). $W$ and $X$ are smoothed to remove outliers (lines 8&9). Then, smoothed $W$ is pseudo-quantized (quantize-then-dequantize to simulate round loss) and compared to the unquantized monitor for quantization error (line 10). This process iterates over various ratios (line 6), selecting the factor with the smallest error as best_s (line 11). Then, this best_s is used to **_Modify the weight_** (line 12), followed by the **_Quant_** (line 13) process using the modified weight.
>
> The drawbacks discussed in lines 146-154 of Section 3.1 also exist for $AWQ_{tweaked}$ in multi-task quantization:
>
> - **_Forward_** (line 1): It can't pass LoRA adapters during activation distribution simulation, causing quantization bias during inference.
> - **_Aggregate Info_**: It uses $X_{mean}=X.mean(0)$, a naive mixed average of multi-tasks' info. Since each task affects each channel differently, simply averaging blurs distributions, ignoring individual effects.
>
> As explained in lines 155-172 of Section 3.1 and shown in Fig 2, MLGPTQ mainly improves the first three steps to tackle GPTQ's issues in multi-task scenarios. Similarly, we can fix AWQ's issues to create a better multi-task quantization algorithm MLAWQ:
>
> - **_Forward_**: MLAWQ loads corresponding LoRA adapter for each task to participate in forward propagation, accurately simulating real activation distribution (line 1).
> - **_Aggregate Info_**: Instead of mixing and averaging features of each column across all tasks to compute $s$, MLAWQ computes the average for each task separately to get $s_i$ (line 3&7). Then it calculates quantization error for each column rather than the entire matrix (line 10). If the $i$-th task results in the smallest quantization error for the $j$-th column, it sets best_s$[j]=s_{i}[j]$ (line 11). This approach allows optimal error minimization, showing each task's individual effect on different channels, enhancing **_Aggregate Info_** (lines 3&6-11), and improving the **_Modify Weight_** (line 12) and **_Quant_** (line 13) processes.
>
> In summary, our work identifies common drawbacks of current single-task quantization methods in multi-task scenarios. By addressing these issues, we can develop more precise multi-task quantization algorithms. Due to time constraints, we couldn't complete the coding and experiments for MLAWQ during rebuttal. However, since the backbone algorithm choice is orthogonal to our work, and GPTQ is popular in LLM, we believe it doesn't diminish our work's significance.
>
> ## Q4
> As discussed in Section 3.3 and _Global Responses_, our multi-task scheduling strategy involves two key techniques: prediction-based SRTF and task grouping. Ablation studies in Figure 9 show that prediction-based SRTF and task grouping increase SLO Attainment by 2.27x and 1.16x, respectively. To address the comment, in _Fig C_ of the PDF, we conducted two more experiments:
>
> - We considered a variant of LoRA-Inlaid, enabling multi-task scheduling strategy while disabling multi-task quantization (i.e., served model not quantized), denoted as "Ours (w/o quant)." Results show "Ours (w/o quant)" increases SLO attainment by 16% compared to S-LoRA, showing the power of our scheduling strategy.
> - We considered a variant of LoRA-Inlaid without output length prediction (instead, using averaged output length of all requests in the same task for sorting), denoted as "Ours (w/o prediction)." Results indicate 18% and 27% efficiency improvements ("Ours" vs. "Ours (w/o prediction)") for 7B and 13B models, respectively.
>
> ## Q5
> As discussed in Section 5 and Appendix F, our work does not consider task fairness currently and we leave it as a future work. When tasks have different priorities, it becomes a case of weighted fairness. A potential solution is using Weighted Fair Queueing (WFQ) [a]. For example, compute a weighted combination of input and output tokens for each task and scale it by the task's priority factor. This information can be seen as the service amount each task receives. Then, we can integrate WFQ to schedule different tasks.
>
> [a] Parekh and Robert. A Generalized Processor Sharing Approach to Flow Control in Integrated Services Networks: The Single-Node Case.

---

> > ### Comment · Reviewer_gmzG · 2024-08-13
> > **Response to authors' rebuttal**
> >
> > Authors' rebuttal mostly addresses my concerns. I will change the score to "accept".

---

> > > ### Author Response · Authors · 2024-08-13
> > > **Thanks for your reply**
> > >
> > > Many thanks for your acknowledgement! Your insightful comments guide significant enhancement to our paper.

---

### Official Review · Reviewer_RQ33 · 2024-07-08

**Soundness:** 3
**Presentation:** 3
**Contribution:** 3
**Rating:** 6
**Confidence:** 3

**Summary:**

The paper addresses the need for efficient fine-tuning and deployment of large language models (LLMs) in multi-task scenarios, which has been largely overlooked in favor of single-task scenarios. Existing quantization methods, such as GPTQ and AWQ, and parameter-efficient fine-tuning techniques like LoRA, are commonly used, but they do not support the integration of multiple tasks due to their limitations in sharing the base model across tasks and handling dynamic task addition. To tackle these issues, the authors propose LoRA-Inlaid, a multi-task LLM serving system that combines a flexible and efficient multi-task quantization algorithm with a novel multi-task scheduling strategy. This system significantly reduces memory consumption, supports real-time task addition, and enhances the stability of online services.

The major contributions of this paper are as follows. The authors introduce an innovative multi-task quantization algorithm, MLGPTQ, which enables the joint quantization of models for multiple tasks, allowing a single quantized base model to be shared across tasks and supporting incremental quantization for new tasks. Additionally, they develop a novel scheduling strategy based on output length prediction and grouping, which minimizes memory consumption and reduces memory swapping overhead in multi-task scenarios. These techniques are integrated into the LoRA-Inlaid system, which demonstrates significant performance improvements over existing LLM serving systems, achieving up to 1.58× higher throughput, 1.76× lower average latency, 2× faster job completion time, and 10× better SLO attainment, all while maintaining model quality. Despite some limitations, such as the need for improved detection of malicious tasks and fairness considerations among tasks, LoRA-Inlaid represents a significant advancement in multi-task LLM serving, highlighting its potential for resource-constrained environments.

**Strengths:**

Originality: The paper introduces an approach to addressing the overlooked area of multi-task fine-tuning and deployment of LLMs. By proposing LoRA-Inlaid, the authors present an innovative multi-task quantization algorithm (MLGPTQ) that allows for joint quantization of models across multiple tasks, supporting incremental quantization for new tasks. This results in their novel multi-task scheduling strategy, which efficiently manages memory consumption and task addition, marking a departure from existing single-task-focused methods.

Quality: The research is evaluated through comprehensive experiments and performance evaluations. The authors provide evidence of significant improvements over existing LLM serving systems, including up to 1.58× higher throughput, 1.76× lower average latency, 2× faster job completion time, and 10× better Service Level Objective (SLO) attainment.

Clarity: The paper is well-structured and clearly presents the problem, methodology, and results. The authors provide detailed explanations of their innovative multi-task quantization algorithm and scheduling strategy, making complex concepts accessible. The use of figures and tables to illustrate performance improvements enhances the clarity and readability of the paper.

Significance: This work addresses a critical gap in the efficient deployment of LLMs in multi-task scenarios. The LoRA-Inlaid system has the potential to greatly improve the efficiency and stability of online services, particularly in resource-constrained environments. The advancements presented in this paper can help advance deployment and scalability of LLMs across various applications, making it an important contribution to the field.

**Weaknesses:**

Editorial comments: Abstract: Avoid excessive use of undefined acronyms in the abstract. Can you explain in the abstract briefly how is LoRA-Inlaid related to MLGPTQ? References: Capitalize proper names and acronyms properly in the References. Some references are incomplete — their publication details (or URL) are missing (e.g., [10])

Outliers: (Figure 3) There are a significant number of outliers represented by circles, especially in the French-English Translation and Table Summary tasks. It might be useful to provide a brief explanation or context for these outliers.
Y-Axis Scale: The y-axis scale goes up to 4000 tokens, but most data points fall well below this range. This could make it harder to see differences in distributions for tasks with shorter lengths. Using a logarithmic scale or breaking the y-axis into two parts could provide better clarity.

Outliers and Variability (Figure 4): There seems to be significant variability in the number of tasks for Skip-join MLFQ (FastServe) and FIFO (S-LoRA). Providing statistical summaries such as mean or median lines could help interpret the data more effectively. Including error bars or confidence intervals would give a better understanding of the variability and reliability of the scheduling strategies.

Malicious Task Detection: The paper acknowledges the need for improved detection of malicious tasks but does not provide detailed solutions or strategies to address this issue. This represents a potential vulnerability in the proposed system that could be exploited in practical applications.

Fairness Considerations: Fairness among tasks is briefly mentioned as a limitation, but the paper lacks an in-depth discussion on how fairness is measured and what specific strategies could be employed to ensure equitable resource distribution among tasks. This is crucial for practical deployment in environments where multiple users or tasks compete for limited resources.

Experimental Scope: While the results are promising, the scope of the experiments is somewhat limited. The paper would benefit from additional experiments across a wider range of tasks and more diverse datasets to fully validate the generalizability and robustness of the proposed system.

Incremental Quantization Details: The paper does not provide extensive details on the incremental quantization process. More information on how new tasks are integrated into the system and the potential impact on existing tasks would strengthen the paper.

**Questions:**

Detection of Malicious Tasks: Can you provide more details on how you plan to enhance the detection of malicious tasks? Are there any preliminary strategies or methods you are considering to address this vulnerability?

Fairness Among Tasks: How do you measure fairness among tasks in the LoRA-Inlaid system? Can you elaborate on the strategies you plan to implement to ensure equitable resource distribution?

Incremental Quantization Process: Could you provide more information on the incremental quantization process? How are new tasks integrated into the system without impacting the performance of existing tasks?

Generalizability of Results: The experiments demonstrate significant improvements, but are these results consistent across a broader range of tasks and datasets? Can you share any additional experimental results or plans for future testing?

Scalability and Real-Time Performance: How does the system scale with a large number of tasks added in real-time? Are there any performance benchmarks or case studies that illustrate the system’s scalability in real-world scenarios?

**Limitations:**

The authors have identified some limitations, such as the need for improved detection of malicious tasks and fairness considerations among tasks. However, these limitations are not fully addressed in the paper.

Addressing Malicious Task Detection: The paper mentions the need for better detection of malicious tasks but does not provide concrete strategies. Constructive suggestion: Develop and describe specific methods for detecting and mitigating malicious tasks, potentially through anomaly detection techniques or secure task validation protocols.

Fairness Among Tasks: Fairness is noted as a limitation, but there is little discussion on how to ensure it. Constructive suggestion: Elaborate on fairness metrics and propose strategies for fair resource allocation among tasks, possibly through dynamic scheduling algorithms that prioritize based on task urgency and resource consumption.

---

> ### Author Rebuttal · Authors · 2024-08-07
>
> ## Editorial Comments
> **Undefined Acronyms**
>
> There are three undefined acronyms in our abstract:
> - **LoRA**, short for Low-Rank Adaptation, is one of the most widely used parameter-efficient fine-tuning techniques for LLMs.
> - **GPTQ** and **AWQ** are state-of-the-art quantization algorithms for LLMs. GPTQ (Frantar et al., ICLR 2023) merges the name of GPT model family with "Q"uantization. AWQ (Lin et al., MLSys 2024) stands for Activation-aware Weight Quantization.
>
> **How is LoRA-Inlaid related to MLGPTQ?**
>
> LoRA-Inlaid consists of two major techniques: the multi-task quantization algorithm and the multi-task scheduling algorithm. We term the multi-task quantization algorithm as MLGPTQ (Multi-LoRA GPTQ). MLGPTQ supports joint quantization of multiple tasks and allows the quantized model to be shared across tasks. It also supports incremental quantization, facilitating dynamic task addition without impacting performance. We'll add the name of our MLGPTQ method to the abstract for clarity.
>
> **References**
>
> Thanks for pointing out the issue. We will update the references in our manuscript accordingly.
> ## Outliers and Variability
> **Explanation for the Outliers in Fig 3**
>
> The outliers are due to the long-tail property of sequence lengths, where few sequences are significantly longer than others. This property is common in NLP datasets (e.g., see section 2 of this article: https://www.harmdevries.com/post/context-length/).
>
> **Scale of the y-axis in Fig 3**
>
> Thanks for the suggestion. We used the same y-axis value range in Fig 3 to clarify differences in sequence length distributions across tasks. For instance, it's clear that the table summary task has shorter output lengths than the code generation task. Thus, we are afraid that using a logarithmic scale or breaking the y-axis into two parts may not achieve clarity as expected.
>
> To address the concern, we have redrawn Fig 3 as suggested, shown in _Fig B_ of the PDF. We will update our manuscript if the reviewer prefers the new figure.
>
> **Variability in Fig 4**
>
> Thanks again for the suggestion. We present the mean and std-dev for each strategy in Fig 4 below. As discussed in Section 3.3, FIFO (S-LoRA) and Skip-join MLFQ (FastServe) lack consideration of tasks scheduled in each step, leading to significant variability. In contrast, our approach shows a smaller std-dev, so our strategy is more suitable for multi-task scheduling.
>
> |  | Ours | FIFO (S-LoRA) | Skip-join MLFQ (FastServe) |
> | --- | --- | --- | --- |
> | mean | 10.11 | 22.22 | 31.49 |
> | std-dev | 1.57 | 2.07 | 4.27 |
>
>
> ## Malicious Task Detection
> One of the most typical use cases of LoRA-Inlaid is the personalization of LLMs, where clients can upload their data to create personalized LoRA adapters using the same base model (or directly upload their self-tuned LoRA adapters). The server is responsible for serving requests from all these clients using the proposed LoRA-Inlaid system. Fortunately, these LoRA adapters are independently manufactured, so we can apply malicious detection to them individually. For instance, the server can prepare a rich set of evaluations to assess the security risks of each LoRA adapter, including violence, discrimination, unlawful responses, etc. If any LoRA adapters fail to pass the evaluation, the server can reject serving them.
>
> ## Fairness Considerations
> To measure fairness among tasks, we can compute a weighted combination of numbers of input and output tokens for each task. This is because the prefilling and decoding phases in LLM inference have different workload characteristics [a] (also why these tokens differ in online service pricing). Then, we can borrow the idea of Weighted Fair Queueing (WFQ) [b] for scheduling different tasks.
>
> [a] Hu et al. Inference without Interference: Disaggregate LLM Inference for Mixed Downstream Workloads.
>
> [b] Parekh and Robert. A Generalized Processor Sharing Approach to Flow Control in Integrated Services Networks: The Single-Node Case.
>
> ## Experimental Scope/Generalizability of Results
> We evaluated our work's effectiveness with 9 datasets from 4 task types, as detailed in Appendix D of our manuscript. We believe these datasets are with significant diversity:
>
> - We considered six languages (French, Czech, Indonesian, Vietnamese, Danish, and Swedish) for the machine translation task, covering various language families.
> - Beyond texts, we also considered two other kinds of inputs (tables and codes) to evaluate our work over diverse tasks.
>
> To address reviewer concerns, we added three datasets with distinct tasks to enhance experimental diversity: grade school math problems (GSM8K), medical QA (Medical_MMLU), and anomaly detection (malicious-600k). As shown in _Table A_ of the one-page PDF, our work consistently outperforms baselines across all tasks. Thus, we believe our work is effective and robust across a wide range of tasks.
>
> GSM8K: https://huggingface.co/datasets/openai/gsm8k
>
> Medical_MMLU: https://huggingface.co/datasets/medalpaca/medical_meadow_mmmlu
>
> malicious-600k: https://huggingface.co/datasets/bgspaditya/malicious-600k
>
> ## Scalability and Real-Time Performance
> In Table 2 of our manuscript, we evaluated the scalability of our work under different numbers of tasks. The results show a small decrease in LoRA-Inlaid throughput (less than 10%) when tasks increase from 2 to 100, even under three request rate levels. To address the comment, we further increase the number of tasks to 1000, and the throughputs under request rates of 5, 10, 20 are 3.42, 4.02, and 4.22 reqs/s, respectively, which are even better than that of S-LoRA serving only 2 tasks. Thus, we believe LoRA-Inlaid has a sound scalability to support both small and large scale workloads in real-world scenarios.
>
> ## Details of the Incremental Quantization Process
> Please refer to the _Global Responses_.

---

> > ### Comment · Reviewer_RQ33 · 2024-08-11
> >
> > Thanks for your clarifications. I acknowledge that I have read these comments in the rebuttal in response to my comments and that I have considered these in my review scores.

---

> > > ### Author Response · Authors · 2024-08-11
> > > **Thanks for your reply**
> > >
> > > We are grateful for your acknowledgement in our response and we believe our paper will be substantially improved by addressing your constructive comments.

---

### Official Review · Reviewer_v1aq · 2024-07-11

**Soundness:** 3
**Presentation:** 3
**Contribution:** 2
**Rating:** 6
**Confidence:** 5

**Summary:**

This paper introduces LoRA-Inlaid, an innovative and efficient system for quantizing and serving Large Language Models (LLMs) in multi-task environments. By utilizing the Multi-LoRA GPTQ (MLGPTQ) algorithm, LoRA-Inlaid facilitates sharing a unified quantized model across various LoRA adapters, significantly reducing memory usage for model deployment. The platform also features a dynamic task addition mechanism that enhances the stability and reliability of online services. Moreover, it introduces a novel multi-task scheduling approach guided by predicted output lengths and task grouping, significantly reducing memory consumption and increasing overall system efficiency. Experimental results demonstrate that LoRA-Inlaid outperforms current state-of-the-art (SOTA) LLM serving systems in terms of throughput, average request latency, Job Completion Times (JCT), and Service Level Objectives (SLO) Attainment without detracting from the model's performance.

**Strengths:**

Originality: The paper introduces LoRA-Inlaid, a novel multi-task serving system for Large Language Models (LLMs), featuring innovations such as multi-task joint quantization, dynamic task addition, and multi-task scheduling. These advancements significantly propel the current field of LLM deployment and services.

Significance: LoRA-Inlaid reduces the memory requirements for model deployment through its multi-task joint quantization and scheduling strategies while maintaining model quality. This is of great importance for resource-constrained environments. Additionally, the addition of dynamic tasks ensures the stability and robustness of online services.

Clarity: The paper has a clear structure and is easy to comprehend, providing detailed procedures of the algorithms and facilitating reproduction. The experiments encompass a variety of metrics, including throughput, average request latency, JCT, and SLO Attainment, demonstrating a well-designed and persuasive set of results.

**Weaknesses:**

1. The paper introduces multi-task quantization, incremental quantization, and predicted output length, which naturally will incur additional computational overhead. However, the paper needs an analysis of the extra costs associated with introducing these techniques.

2. The paper is missing absolute accuracy comparison experiments for the unquantized model. Both Figure 5 and Figure 6 represent relative accuracy drops compared to the quantization model without providing a direct comparison to the baseline performance of the unquantized model.

3. An excessive amount of content from Chapter 3 has been relegated to the appendices, with the main text providing a succinct description of the specific methods and also needing formulas to aid in explanation, which affects the clarity of the paper.

**Questions:**

1. In Figure 5, why is there a lack of error bars for RTN?

2. In Figure 5, why does the accuracy drop for some tasks with 3-bit quantization appear to be less than that for 4-bit quantization?

3. In the three charts of Figure 6, MLGPTQ and GPTQ have their respective strengths and weaknesses across various metrics. It is suggested that additional tasks be added to demonstrate the superiority of MLGPTQ in terms of accuracy.

4. Currently, there are only experiments on accuracy drops. Please supplement with accurate experiments.
In the efficiency tests, only S-LoRA used half-precision, which significantly increased its computational overhead, leading to an unfair comparison experiment.

5. In addition to the End-to-end system performance, could you independently analyze the individual impacts of components such as multi-task quantization, incremental quantization, and predicted output length on each performance of LLM services?

6. Please add formalized descriptions of dynamic task addition and multi-task scheduling in the main text and a brief outline of the specific method.

**Limitations:**

Yes.

---

> ### Author Rebuttal · Authors · 2024-08-07
>
> ## W2 & Q4 (first half)
> Since divergent metrics (SacreBLEU and ROUGE-1) are used for different tasks, Fig 5 shows the relative performance to align the y-axis. We have presented the detailed results in _Table A_ of the one-page PDF, showing that MLGPTQ consistently outperforms the baselines.
> ## Q1
> Thanks for pointing out the issue. The std-dev is also provided in _Table A_. RTN has a lower std-dev than the other quantization approaches. This is because RTN is deterministic while the other approaches introduce randomness as we shuffle the calibration sets for every time of quantization. The randomness in RTN (as well as Unquantized) only comes from the non-deterministic outputs (the LLM may produce different outputs for the same input to enhance creativity), however, such randomness hardly affects the evaluation.
> ## Q2
> Fig 5 uses different value ranges for the y-axes of 3-bit and 4-bit quantization, leading to misunderstanding. _Table A_ shows 3-bit quantization is consistently worse than 4-bit quantization.
> ## Q3
> In Fig 6, GPTQ is the baseline that quantizes the model individually for each task, so the quantized model of GPTQ cannot be shared among different tasks. In contrast, MLGPTQ quantizes the model jointly for all tasks, ensuring the quantized model is shareable. Hence, it is reasonable that GPTQ is better on some metrics. We considered GPTQ as a reference since it fulfills the two key factors (discussed in lines 283-286 of Section 4.2), aiming to better anatomize the effectiveness.
>
> Due to the space constraint, we only provided the results on 3 datasets in Fig 6. To address the comment, we have presented the results on the other 3 datasets of the translation task in _Fig D_ of the one-page PDF (the other tasks are not considered since metrics like G_BLEU, S_BLEU, and NIST_MT do not apply). The results are consistent with Fig 6 --- both MLGPTQ and GPTQ, which fulfill the two factors, outperform the other approaches in almost all metrics. Moreover, since MLGPTQ produces a shareable quantized model while GPTQ cannot, the results verify that MLGPTQ is suitable for multi-task quantization.
> ## Q4 (second half)
> Although the model is quantized, the computation during inference is still executed in half-precision. Before the computation of each layer, the corresponding model weights are temporarily dequantized (e.g., `MatMul4Bit`in `bitsandbytes`: https://github.com/bitsandbytes-foundation/bitsandbytes/blob/0.43.0/bitsandbytes/autograd/_functions.py#L516). Thus, mmodel quantization does not decrease computational overhead but introduces a minor overhead of dequantization.
>
> Besides, S-LoRA does not support deploying quantized models since existing quantization methods do not fit multi-task scenarios, as discussed in Section 3.1 of our manuscript, so we could only use half-precision for S-LoRA. (LoRA-Inlaid is the first system that supports deploying quantized models for multi-task serving.) Thus, we believe the system comparison in our work is fair.
> ## W1 & Q5
> Below we analyze the individual impact of each component.
>
> **Multi-task Quantization**
>
> As discussed in the response to _Q4 (second half)_, moodel quantization does not decrease computational overhead. However, it reduces the memory consumption of storing model weights so that we can preserve more memory for KVCache to improve overall efficiency.
>
> To assess the impact of multi-task quantization, we considered a variant of LoRA-Inlaid, which disables quantization (i.e., the served model is not quantized), denoted as "Ours (w/o quant)" in the right of _Fig C_ of the one-page PDF. The results show that multi-task quantization brings 39% improvement ("Ours" vs. "Ours (w/o quant)") when serving the 7B model, and disabling it leads to OOM when serving the 13B model.
>
> The time cost of quantization is discussed in the analysis of incremental quantization below. Note that we can quantize the model before serving, so it does not affect the performance of serving.
>
> **Incremental Quantization**
>
> The incremental quantization aims to support dynamic task addition without halting the serving. As introduced in our _Global Responses_, there are two steps in quantization, and we avoid redundant computation in the first step of incremental quantization. Meanwhile, a layer-by-layer mechanism is developed to reduce the memory consumption of incremental quantization.
>
> To evaluate its impact, we conducted an experiment where there are 5 tasks in the ongoing service and another 5 tasks need to be added. We measured the time cost of three approaches:
> - Full quantization with 10 tasks, which halts the serving.
> - An offline variant of our incremental quantization with the 5 new tasks, which halts the serving. However, it does not need to perform the layer-by-layer quantization.
> - Our incremental quantization with the 5 new tasks, which works concurrently with the ongoing service.
>
> As shown below, by avoiding the redundant computation, the time cost of the first step can be reduced greatly, accelerating quantization. Moreover, although the layer-by-layer mechanism slows down the quantization by 1.26 times due to the extra IO, it reduces the memory greatly and does not halt the serving.
> ||Step 1|Step 2|Total|Peak Memory (GB)|
> |-|-|-|-|-|
> |Full Quant|1403(±21)s|415(±6)s|1818(±22)s|9.2|
> |Incr Quant (offline)|663(±11)s|416(±5)s|1079(±12)s|9.2|
> |Incr Quant|889(±11)s|469(±6)s|1358(±13)s|2.5|
>
> **Output Length Prediction**
>
> As introduced in our _Global Responses_, the output length prediction is done on CPU and overlaps with the LLM inference on GPU. To measure its impact, we experimented with a variant of LoRA-Inlaid without the output length prediction (instead, the averaged output length of each task is used), denoted as "Ours (w/o prediction)" in the left of _Fig C_ of the one-page PDF. The results show that it brings 18% and 27% improvement ("Ours" vs. "Ours (w/o prediction)") for the 7B and 13B models.
> ## W3 & Q6
> Please refer to the _Global Responses._

---

> > ### Comment · Reviewer_v1aq · 2024-08-13
> >
> > Thanks for the authors' detailed feedback. It has addressed my concerns. In light of these explanations, I will revise my score accordingly.

---

> > > ### Author Response · Authors · 2024-08-13
> > > **Thanks for your reply**
> > >
> > > Thank you for your time and consideration! Your detailed comments are extremely helpful, and we believe addressing your comments will significantly improve our paper.

---

### Author Rebuttal · Authors · 2024-08-07

# Global Responses

We are grateful to all reviewers for the careful reviews. We provide _Global Responses_ to common questions, followed by individual responses. Please refer to the attached one-page PDF for related figures and tables.

## Details of the Incremental Quantization and Multi-task Scheduling

We acknowledge the oversight that these two modules are not introduced comprehensively in the main text of our manuscript due to the space constraint. To address the reviewers' concerns, we would like to elaborate below.

### **Incremental Quantization**

In Appendix A.2 of our manuscript, we provided the full quantization process of MLGPTQ in Alg 1. To quantize each model weight $W$ with $T$ tasks, there are two steps:

- [Lines 1-2 of Alg 1] Compute the Hessian matrices for all tasks (i.e., compute $H_1^{-1},H_2^{-1},\cdots,H_T^{-1}$).
- [Lines 3-13 of Alg 1] Max-aggregate these $T$ Hessian matrices (i.e., $H_{tmp}=MaxAgg(H_1^{-1},\cdots,H_T^{-1})$)  and update the model weight.

When there are new tasks, a naive solution is to perform full quantization again. Denote $T_1,T_2$ as numbers of existing and new tasks, respectively. The naive solution runs the two steps above with $T=T_1+T_2$. However, this leads to **redudant computation** of $H_1^{-1}, H_2^{-1}, \cdots, H_{T_1}^{-1}$. Furthermore, given the **commutative property of the max-aggregation operation**, we have

$MaxAgg(H_1^{-1},\cdots, H_{T_1+T_2}^{-1})=MaxAgg(MaxAgg(H_1^{-1},\cdots, H_{T_1}^{-1}),MaxAgg(H_{T_1+1}^{-1},\cdots,H_{T_1+T_2}^{-1})),$

where the first term $MaxAgg(H_1^{-1},\cdots,H_{T_1}^{-1})$has already been computed as $H_{tmp}$ in the previous quantization.

Inspired by this, we can cache $H_{tmp}$, so the incremental quantization can be done as follows:

- [Same as lines 1-2 of Alg 1] Compute the Hessian matrices for new tasks $H_{T_1+1}^{-1},\cdots,H_{T_1+T_2}^{-1}$.
- [Same as lines 3-13 of Alg 1] Max-aggregate the $T_2+1$ matrices (i.e., $H_{T_1+1}^{-1},\cdots,H_{T_1+T_2}^{-1}$ and the cached $H_{tmp}^{(cached)}$) and update the model weight.

Obviously, incremental quantization with $T_2$ tasks is identical to full quantization with $T_1+T_2$ tasks, while **avoiding redundant computation**.

To avoid halting the ongoing services, LoRA-Inlaid spawns a background thread for incremental quantization. In addition, to reduce the memory consumption of incremental quantization, it is done in a **layer-by-layer** manner: for each (unquantized) model weight, we load it from CPU memory to GPU memory, perform incremental quantization, remove it from GPU memory, and proceed to the next model weight. The IO between CPU-GPU is overlapped with computation. Thus, LoRA-Inlaid supports seamless task addition on the fly and has very little influence on the ongoing services.

### **Multi-task Scheduling**

In Appendix B of our manuscript, we illustrated multi-task scheduling in Alg 2~4 and discussed its workflow in lines 529-542. For better understanding, we included a flow chart in _Fig A_ of the one-page PDF.

Given the prompt of a request, there are two phases for LLM inference: _the prefilling phase_ takes the prompt to compute the key-value (KV) cache and generates the first output token in a single step, and _the decoding phase_ takes the last generated token and KV cache to generate subsequent tokens. The decoding phase needs to be executed for multiple steps for each request, with each step generating only one token, until the EOS token is generated.

In the field of LLM serving, the scheduling strategy is responsible for determining what should be done in each scheduling step. There are three choices for each step:

- [lines 7-11&16-20 of Alg 2] generate a batch from the prefilling queues;
- [line 22-25 of Alg 2] schedule a batch from the decoding queues;
- [lines 27-28 of Alg 2] continue decoding for the current batch.

The determination in our work follows the standard rules in the field of LLM serving (e.g., switch to prefilling if we have done decoding for several consecutive steps).

To enhance multi-task scheduling, there are two key techniques in our work (i.e., the two solutions in Section 3.3) when generating/scheduling a batch.

**Scheduling Guided by Output Length Prediction**: Considering the sequence length variation across tasks, we take the remaining output length information into account.
-  [lines 214-216 of Section 3.3] Upon receiving a new request, we predict its output length on CPU using a small model (255MB) and enqueue the request into the _prefill_reqs_. Note that the output length prediction takes about 16ms for one request on CPU, while it takes about 200ms to finish the inference of one request on GPU. Hence, we can completely overlap the prediction, without occupying any GPU computing resources.
-  [lines 2-3 of Alg 3 & lines 2-3 of Alg 4, also in lines 217-219 of Section 3.3] For each scheduling step, we sort the queues to achieve the Shortest Remaining Time First (SRTF).

**Reducing Tasks Involved via Grouping**: To avoid expensive memory access overhead in each step, there are two efforts.
- [lines 6&11 of Alg 3, also in lines 233-234 of Section 3.3] We restrict the number of involved tasks below a threshold (i.e., $\beta$ in Section 3.3) when generating a new batch from the prefilling queues;
- [lines 6&11 of Alg 4, also in lines 235-236 of Section 3.3] We prioritize tasks involved in the previous step to avoid swapping LoRA adapters frequently.

Besides, our work also maintains the waiting time of each request to avoid starvation (i.e., by the hungry queues in Alg 2-4), which is common in LLM serving.

To conclude, the scheduling in our work is not complex. Compared with existing scheduling strategies, we leverage two techniques considering the characteristics of multi-task serving. Thus, in our manuscript, we focused on the two techniques in Section 3.3, while the detailed routine (e.g., the determination of the three choices for each step) is deferred to the appendix.

---

### Decision · Program_Chairs · 2024-09-25

**Decision:**

Accept (poster)

**Comment:**

The paper introduces LoRA-Inlaid, an innovative multi-task large language model serving system that optimizes memory usage and supports dynamic task addition through novel quantization and scheduling algorithms. Reviewers praised its practical impact and theoretical contributions but raised concerns about computational overhead, clarity, and limited empirical validation. Authors responded with detailed clarifications and additional experiments, effectively addressing most concerns. The reviewers' consensus was positive, recommending acceptance due to the system's significant advancements in LLM deployment.  The AC concurs with this recommendation and the paper may be accepted to NeurIPS.